# CACR: REINFORCING TEMPORAL ANSWER GROUNDING IN VIDEOS VIA CANDIDATE-AWARE CAUSAL REASONING

## ABSTRACT

The growing need for direct answer retrieval from videos underscores the importance of Temporal Answer Grounding in Videos (TAGV)—the task of localizing the specific video segment that answers a natural language query. TAGV remains challenging, as it requires understanding semantically complex questions and handling the extreme length disparity between untrimmed videos and short answer segments. Current methods often underperform due to sensitivity to redundant content or limited visual reasoning ability. To overcome these issues, we introduce a Candidate-Aware Causal Reasoning (CACR) framework. Our approach first employs a Visual-Language Pre-trained (VLP) model to efficiently generate K candidate segments, then applies a temporal logic reasoning module strengthened by a rejection reward mechanism and optimized through Group Relative Policy Optimization (GRPO) for robust inference. Extensive experiments on four benchmarks show that our method achieves state-of-the-art performance in mean Intersection-over-Union (mIoU), offering a new direction for reasoning-based retrieval in long videos.We also publish our code at:https://github.com/anonymous1118-10/opencode-CACR.

## 1 INTRODUCTION

The rapid growth of video data has increased the demand for methods that can directly retrieve answers from videos in applications such as tutorials, fault diagnosis, and event reconstruction. In these scenarios, temporal continuity and relational dynamics are often more important than static images or text, making Temporal Answer Grounding in Videos (TAGV) a key research focus in AI and computer vision. Unlike traditional Visual Question Answering (VQA), which outputs text, or Temporal Sentence Grounding in Videos (TSGV), TAGV deals with semantically complex queries that often depend on visual demonstrations rather than pure language. This leads to a significant semantic gap and higher task complexity Li et al. (2024). An additional challenge in long videos is the extreme ratio between the total video length and the duration of the target answer segment. As shown in Fig. 1A(a), the TutorialVQA dataset exhibits an extreme ratio of 48.9—meaning the model must search through nearly 49 seconds of video to locate a 1-second answer. This highlights the urgent need for efficient long-video search mechanisms and fine-grained temporal localization.

Current approaches to TAGV suffer from several key limitations. Methods based on VLP models Zhang et al. (2021; 2020a); Li et al. (2024); Weng & Li (2023) rely on low-level feature matching, making them sensitive to dynamic scenes and redundant content, and they often lack deeper relational reasoning. Self-supervised step segmentation methodsDvornik et al. (2023); Tang et al. (2019) may miss subtle visual transitions and semantic relationships. Text-driven LLM-based approaches OpenAI (2023); Driess et al. (2023) are constrained by subtitle quality and weak visual integration. While recent reinforcement learning-based LVLM methods Guo et al. (2024; 2025b); Yang et al. (2023); Zhao et al. (2025) frequently lead to overfitting and poor task adaptation. Recently, reinforcement learning-based LVLM methods Wang et al. (2025b) show promise by directly optimizing metrics like IoU, they often process the entire video, leading to information loss and reasoning bias when the target segment is very short.

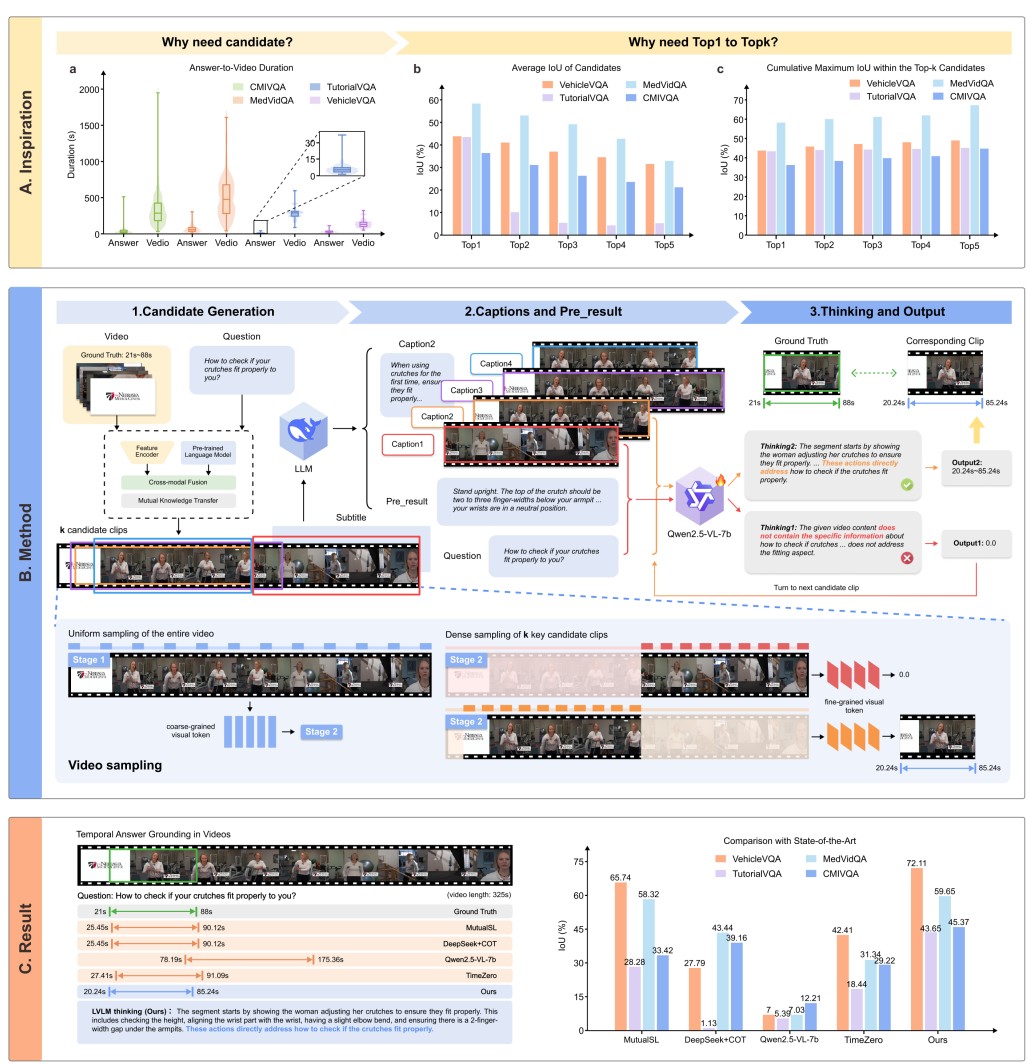

Figure 1: Overview of the proposed CACR framework. Fig. 1A illustrates the inspiration, showing the extreme length contrast between answer and video segments (a), the average IoU of candidates at different ranks (b), and the cumulative maximum IoU within top-K selections (c). Fig. 1B outlines the method pipeline, including candidate generation (B.1), caption and answer hypothesis extraction (B.2), and causal reasoning for final prediction (B.3). Fig. 1C compares the performance of our approach with state-of-the-art methods.

To tackle the core issue of extreme length disparity, we propose to narrow the processing scope for the subsequent reasoning model. We first employ a VLP-based candidate selection (VBCS) algorithm to efficiently generate a manageable set of K high-quality candidate segments from the long video. This coarse screening focuses the model's attention on promising regions, alleviating information overload. Crucially, we observed that the maximum IoU between the top-K candidate segments and the ground truth increases monotonically with K (Fig. 1A(b,c)). This finding indicates that even if the single best candidate is imperfect, a larger, high-quality candidate pool is likely to contain a segment very close to the ground truth. Inspired by this, we introduce the Candidate-Aware Causal Reasoning (CACR) framework.

Within the refined candidate set, our framework employs a sophisticated temporal logic reasoning module. This module is strengthened by a rejection reward mechanism, encouraging the model to discard incorrect candidates when uncertain, thereby improving robustness. The reasoning process is optimized using Grrop Relative Policy Optimization (GRPO), which enhances policy learning by

comparing groups of responses. To further aid reasoning, we integrate prior knowledge through an LLM-driven answer hypothesis generator and a subtitle summarization module, improving semantic alignment between the query and video content.Our main contributions are as follows.

(i) We propose the VBCS algorithm to efficiently generate high-quality candidate segments, mitigating the information overload in long videos.

(ii) We design a temporal logic reasoning module reinforced by a rejection reward mechanism and optimized via GRPO, enhancing distractor discrimination and causal reasoning capability.

(iii) Extensive experiments on four benchmarks demonstrate that our method achieves state-of-the-art performance in mIoU, showcasing its strong generalization ability.

## 2 RELATED WORK

**Temporal Answer Grounding in Videos (TAGV)** requires a model to localize the precise temporal segment in an untrimmed video that directly answers a natural language question. This task demands more sophisticated multimodal reasoning compared to Temporal Sentence Grounding in Videos (TSGV). Early approaches often followed a two-stage retrieve-and-verify paradigm (Zhang et al., 2020a;b), which first generated candidate segments and then performed fine-grained matching. Subsequent sliding-window Transformers (Lei et al., 2021a; Qu et al., 2022) framed TAGV as a set prediction task, directly regressing segment boundaries. Methods built on the pre-train then fine-tune paradigm (Lei et al., 2021b; Yan et al., 2022) leveraged large-scale image-text pre-training before adapting to the task. While these methods have improved benchmark performance, they suffer from several limitations, including shallow cross-modal interaction, reliance on predefined windows or large-scale image-text pairs, and a fundamental misalignment with TAGV's requirement for precise temporal localization—often resulting in temporal-semantic mismatches. More recent caption-based approaches, such as VTPSL (Li et al., 2024) and MutualSL (Weng & Li, 2023), enhance cross-modal alignment using video captions and pre-trained language models. However, they still lack the ability to model inter-step relationships and remain sensitive to redundant content (e.g., presenter narration), leading to localization bias.

**Reasoning in Large Vision-Language Models**. Recent studies have increasingly adopted end-to-end frame-based methods that fine-tune LVLMs through supervised fine-tuning (SFT) with autoregressive losses Ren et al. (2024); Zeng et al. (2025); Hannan et al. (2025); Wu et al. (2025); Guo et al. (2025b); Yang et al. (2023); Zhao et al. (2025). However, these methods often underperform compared to feature-based approaches. We argue that a key reason for LVLMs' suboptimal performance in temporal grounding lies in the excessive penalty imposed on false negatives during SFT. For example, in an action localization task where the ground truth spans frames 10–30, a model predicting frames 9–29 may still capture the essential action with only minor boundary shifts—well within human perceptual tolerance. Yet, autoregressive loss heavily penalizes such reasonable predictions, leading to overfitting and limited generalization. Inspired by recent breakthroughs in reinforcement learning (RL) for post-training large language models—such as GPT-4o1 OpenAI (2023), DeepSeek-R1 Guo et al. (2025a), Kimi-K1.5 Team et al. (2025), Qwen-3 Yang et al. (2025), and Magistral Mistral-AI et al. (2025)—where methods like GRPO, RLVR, and Expert Iteration have advanced the state-of-the-art in code and mathematical reasoning, and encouraged by their recent extension to long-video reasoning in GRPO-enhanced multimodal LLMs Zhang et al. (2025), we explore whether GRPO can serve as an effective alternative for TAGV.

**Group Relative Policy Optimization** is an emerging reinforcement learning technique that optimizes policy gradients by computing relative advantages within sample groups, thereby substantially reducing training variance and enhancing decision quality. Initially successful in text-based domains such as mathematics and programming OpenAI (2023); Guo et al. (2025a), GRPO has demonstrated strong reasoning capabilities, enabling large models to approach human-level performance on complex tasks Evstafev (2025); OpenAI (2023). Recent studies have extended GRPO to video structure understanding. Works such as Ge et al. (2025); Wang et al. (2025a;b) show that reinforcement learning-based post-training can significantly improve model reasoning in video-related tasks. However, these methods typically process the entire video without accounting for the extreme timespan ratio between the answer segment and the full video duration, often leading to substantial information loss and biased results.

## 3 METHOD

The TAGV task aims to temporally locate the video segment corresponding to a given text query in a long video. A video is represented as a sequence of frames $v_1, \ldots, v_T$, the language query is $Q$, and the target segment is defined by its temporal boundaries $[t_s, t_e]$, where $t_s, t_e \in \mathbb{R}^+$ (in seconds).

We introduce CACR, a framework designed to enhance the capability of LVLMs for the TAGV task using a reinforcement learning approach. Unlike traditional RL pipelines that require a supervised fine-tuning (SFT) phase for cold-start initialization, CACR directly employs Group Relative Policy Optimization (GRPO) without explicit pre-training. This design choice is motivated by the strong inherent capabilities of our base model, Qwen2.5-VL-7B-Instruct, which already possesses robust instruction-following and reasoning abilities. The model's pre-existing proficiency in understanding temporal concepts and following structured output formats provides a sufficient foundation for effective policy optimization through reward signals alone. First, in Section 3.1, we introduce the VBCS framework for generating high-quality candidate segments. Then, in Section 3.2, we describe the reinforcement learning training mechanism of CACR.

### 3.1 TOP-K CANDIDATE GENERATION VIA VBCS

To address the length disparity between answer segments and full videos, we leverage the Visual-Language Pre-training based Candidate Selection (VBCS) framework to generate high-quality top-$K$ candidate segments. VBCS is designed as a flexible framework that can incorporate any state-of-the-art VLP based localizer. In this work, we instantiate VBCS with the MutualSL(Weng & Li, 2023), which currently achieves top performance among open-source models for temporal grounding, to ensure precise cross-modal alignment.

Given an untrimmed video $V = \{v_i\}_{i=1}^T$, subtitle sequence $S = \{(t_j^{\text{start}}, t_j^{\text{end}}, s_j)\}_{j=1}^m$ (where each tuple contains start/end times in seconds and text content), and query $Q$, VBCS outputs $K$ candidate segments $C_{\text{vis}} = \{(c_k^s, c_k^e)\}_{k=1}^K$ representing start/end times in seconds. These candidates aim to cover the ground-truth segment $[V_s^*, V_e^*]$ (with start/end times in seconds). The workflow of VBCS (instantiated with MutualSL) is formalized in Algorithm 1.

---

**Algorithm 1** Top-$K$ Candidate Generation via VBCS

---

**Require:** Untrimmed video $V$, subtitles $S$, query $Q$, candidate number $K$, temporal extension $\Delta t$
**Ensure:** Top-$K$ candidate segments $C_{\text{vis}} = \{(c_k^s, c_k^e)\}_{k=1}^K$
  1: **Stage 1: Cross-Modal Feature Extraction**
  2: Extract visual features: $\mathbf{V} = \text{I3D}(V)$
  3: Extract textual features: $\mathbf{T} = \text{PLM}([Q; S])$
  4: Fuse features using Context-Query Attention
  5: **Stage 2: Dual-Predictor Probability Estimation**
  6: Visual predictor: estimate frame-level start/end probabilities $V_s, V_e$
  7: Textual predictor: estimate token-level start/end probabilities $T_s, T_e$
  8: **Stage 3: Cross-Modal Alignment (Training)**
  9: Align predictions using timeline mapping $\mathbb{Q}(\cdot)$
 10: Train with combined loss: base prediction loss + mutual transfer loss
 11: **Stage 4: Top-$K$ Candidate Selection (Inference)**
 12: Generate candidate segments by pairing top-ranked start/end positions:
 13: Candidates $= \{(\max(0, v_s - \Delta t), \min(v_e + \Delta t, T)) \mid v_s \in \text{Top-}K(V_s), v_e \in \text{Top-}K(V_e), v_s < v_e\}$
 14: Select first $K$ candidates in generation order
 15: **return** $C_{\text{vis}}$

---

The VBCS framework, instantiated with MutualSL, operates through four sequential stages (Algorithm 1): (1) cross-modal feature extraction and fusion, (2) dual-predictor probability estimation, (3) cross-modal alignment through mutual knowledge transfer, and (4) temporally-extended candidate generation and selection. This approach addresses the length disparity challenge by narrowing the processing scope for subsequent modules while maintaining a high recall of the ground-truth segments.

## 3.2 GRPO-BASED TEMPORAL LOCALIZATION FRAMEWORK

After obtaining the top-$K$ candidate segments $C_{\text{vis}} = \{(t_{s_k}, t_{e_k})\}_{k=1}^{K}$ generated by VBCS, we propose **CACR** —a reinforcement learning framework based on GRPO. This framework drives LVLMs to perform deep reasoning over the candidate segments, rather than directly regressing timestamps. Specifically, the model adopts a "reason-first, localize-later" strategy, fully leveraging causal priors and text summaries generated by an LLM to enhance the understanding of step-level logic and semantic context. Furthermore, the reasoning process is optimized via a composite reward function that integrates temporal IoU reward, rejection reward, and template reward, improving the accuracy of temporal boundary prediction and robustness against distracting segments. This approach significantly mitigates localization bias and enhances generalization capability weakened by the lack of causal reasoning.

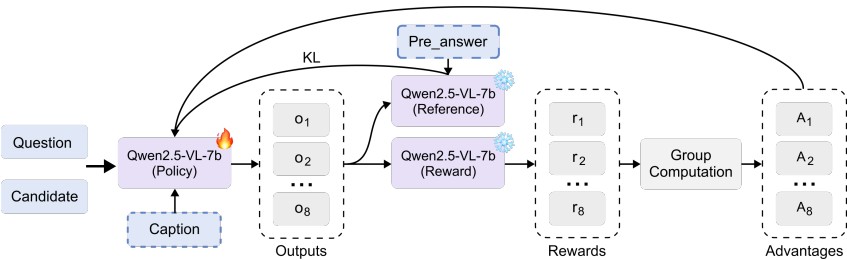

Figure 2: The policy model (Qwen2.5-VL-7B) takes the Question, Candidate segment, and Caption as input to generate multiple outputs ($o_1$–$o_8$). These outputs, along with a Pre-answer, are fed into the reward and reference models to compute rewards ($r_1$–$r_8$), which are then used to calculate advantages ($A_1$–$A_8$). The policy model is subsequently optimized using a KL divergence penalty.

### 3.2.1 GRPO ALGORITHM FOUNDATION

As the core algorithm of our CACR framework, GRPO optimizes the policy model $\pi_\theta$ (i.e., the decision-making policy of the LVLM) through a rule-based reward function. It is particularly suitable for tasks with well-defined output spaces, such as temporal localization. The core mechanism operates as follows: Given an input question $P$, the model generates $G$ candidate responses $\mathbf{o} = \{o_1, \ldots, o_G\}$. A designed reward function $r(\cdot)$ computes the rewards $\{r(o_1), \ldots, r(o_G)\}$. The objective of GRPO is to maximize the weighted sum of rewards:

$$R(\mathbf{o}) = \sum_{i=1}^{G} \frac{\pi_\theta(o_i)}{\pi_{\theta_{\text{old}}}(o_i)} \cdot \frac{r(o_i) - \mu_{\mathcal{R}}}{\sigma_{\mathcal{R}}}$$

where - $\mu_{\mathcal{R}} = \text{mean}\left(\{r(o_i)\}_{i=1}^{G}\right)$ is the group mean of the rewards, - $\sigma_{\mathcal{R}} = \text{std}\left(\{r(o_i)\}_{i=1}^{G}\right)$ is the group standard deviation of the rewards, - $\pi_{\theta_{\text{old}}}$ denotes the policy parameters before the latest update. The normalization term $\frac{r(o_i) - \mu_{\mathcal{R}}}{\sigma_{\mathcal{R}}}$ serves as an estimate of the relative advantage of response $o_i$, denoted as $A(o_i)$. A positive $A(o_i)$ indicates that the response's reward is above the group average, while a negative value signifies below-average performance.

### 3.2.2 POLICY OPTIMIZATION OBJECTIVE

As illustrated in Figure 2, the policy optimization objective of GRPO aims to maximize the cumulative reward while constraining the currently trainable policy model $\pi_\theta$ via a KL divergence regularization term, preventing it from deviating excessively from a stable reference model $\pi_{\text{ref}}$. The objective function is defined as follows:

$$\max_{\pi_\theta} \mathbb{E}\left[\sum_{i=1}^{G} \frac{\pi_\theta(o_i)}{\pi_{\text{old}}(o_i)} A(o_i)\right] - \beta \cdot D_{\text{KL}}(\pi_\theta \parallel \pi_{\text{ref}})$$

where: $\beta > 0$ is a regularization coefficient that balances reward optimization and policy conservatism, $A(o_i) = \dfrac{r(o_i) - \mu_{\mathcal{R}}}{\sigma_{\mathcal{R}}}$ denotes the relative advantage of response $o_i$ within the group, $D_{\mathrm{KL}}(\pi_\theta \parallel \pi_{\mathrm{ref}})$ measures the distributional divergence between the current policy and the reference model.

In traditional GRPO methods, both the policy model and the reference model often receive identical input prompts "Based on the caption {caption}, answer the {question}.", making the reference model susceptible to caption noise or local details, thereby providing an unstable or biased regularization baseline. To address this issue, we propose a differentiated prompting strategy: The policy model $\pi_\theta$ receives the raw input state $s = (C_{\mathrm{vis}}, Q, C_{\mathrm{vis_{caption}}})$, which includes video frames, the query, and the candidate segments along with their caption summaries. $C_{\mathrm{vis_{caption}}}$ is generated by LLM and represents a summary based on the captions of the candidate segments. The reference model $\pi_{\mathrm{ref}}$ receives a semantically enhanced and reformulated input state $s' = (C_{\mathrm{vis}}, Q, \text{Pre-answer})$, prompted with a statement like "Consider the following hypothetical answer: Pre-answer". Here Pre-answer is generated by LLM and represents a hypothetical answer to the query. This design provides the reference model with a higher-level semantic context focused on the query's intent.

This design ensures that the KL divergence term $D_{\mathrm{KL}}(\pi_\theta \parallel \pi_{\mathrm{ref}})$ acts not merely as a constraint against an initial policy, but rather guides the policy model towards outputs that are aligned with a more stable and intent-aware prior distribution. This mechanism significantly improves training stability and ensures that while pursuing high rewards, the model's predictions remain strongly correlated with the true semantic intent.

In the temporal localization task, the reward $r(o_i)$ is defined as the composite reward $R_{\mathrm{total}}$ described in Section 3.2.3. The policy $\pi_\theta$ is optimized via gradient ascent, progressively favoring the generation of temporal boundaries that exhibit both high reward and strong semantic consistency.

### 3.2.3 COMPOSITE REWARD FUNCTION

The reward function consists of three components:

$$R_{\mathrm{total}} = R_{\mathrm{fmt}} + (1 - \alpha) \cdot R_{\mathrm{IoU}} + \alpha \cdot R_{\mathrm{rej}} \tag{1}$$

The balance coefficient $\alpha$ is empirically set to 0.8 during training, emphasizing the importance of correct rejection decisions while maintaining strong temporal alignment incentives.

**Template Reward** $R_{\mathrm{fmt}}$ encourages the model to follow structured output format: 1 when format is correct, 0 otherwise.

**Temporal Alignment Reward** $R_{\mathrm{IoU}}$ activates when the model outputs valid boundaries $[t_s^*, t_e^*]$ with $t_s^* > 0, t_e^* > 0$, computing IoU between prediction and ground truth:

$$R_{\mathrm{IoU}} = \frac{|[t_s^*, t_e^*] \cap [t_s^{\mathrm{GT}}, t_e^{\mathrm{GT}}]|}{|[t_s^*, t_e^*] \cup [t_s^{\mathrm{GT}}, t_e^{\mathrm{GT}}]|} \tag{2}$$

**Rejection Reward** $R_{\mathrm{rej}}$ equals 1 when the model outputs $[0.0, 0.0]$ and the candidate segment $\mathcal{C}_k$ has no overlap with ground truth (IoU=0), otherwise 0.

### 3.3 INFERENCE PIPELINE

During inference, the pipeline processes an input video through three main stages to produce the final temporal prediction. The specific steps are as follows:

**Stage 1: Candidate Proposal Generation** The input video is processed by the VBCS, which predicts $k$ candidate segments based on an analysis of video content features. These segments represent temporal intervals likely to contain relevant information, forming a preliminary set of temporal proposals.

**Stage 2: Caption and Pre-Answer Generation** For each candidate segment, the corresponding video clip along with the question $q$ is input into the LLM model to generate a descriptive caption and a preliminary answer (pre-answer). These outputs provide enriched semantic context to support fine-grained reasoning in the subsequent stage.

**Stage 3: Iterative Temporal Reasoning and Validation** The $k$ candidate segments, along with their corresponding captions and pre-answers, are sequentially fed into the CACR model in the order provided by VBCS. For each candidate, CACR performs cross-modal reasoning based on the provided information and outputs a timestamp $[t_s^*, t_e^*]$. The validity of the timestamp is verified by checking whether $t_s^* > 0$ and $t_e^* > 0$. If the timestamp is valid, it is immediately returned as the final prediction. If the output is $[0, 0]$, indicating rejection, the model proceeds to evaluate the next candidate in the sequence. This process iterates until the first valid timestamp is produced.

# 4 EXPERIMENTS

## 4.1 EXPERIMENTAL SETTINGS

**Datasets.** We conduct a comprehensive evaluation on four challenging instructional video datasets for TAGV: VehicleVQALuo et al. (2019), TutorialVQAColas et al. (2019), MedVidQAGupta et al. (2023), and CMIVQALi et al. (2025). These datasets span automotive, software editing, and medical domains, covering diverse application scenarios to comprehensively test model robustness and generalization capabilities. VehicleVQA exhibits the smallest Dur./Span (5.93), indicating relatively coarse-grained temporal reasoning requirements. In contrast, TutorialVQA presents an extreme ratio of 48.9, requiring models to locate 1-second answer segments within 49-second videos on average, which demands highly efficient long-sequence search and fine-grained temporal localization. Table 1 summarizes the dataset statistics.

Table 1: Statistics of the instructional video datasets used in our experiments. Duration values are reported in seconds.

| Dataset | Domain | Videos | QA Pairs | Duration Range (s) | Avg. Dur. (s) | Avg. Span (s) | Dur./Span |
|---|---|---|---|---|---|---|---|
| VehicleVQA | Automotive | 107 | 8,632 | 38.87–309.06 | 125.27 | 21.12 | 5.93 |
| TutorialVQA | Software Editing | 76 | 6,195 | 83.40–588.35 | 284.12 | 5.81 | 48.90 |
| MedVidQA | Medical | 899 | 3,010 | 32.99–1,596.68 | 415.99 | 61.97 | 6.71 |
| CMIVQA | Medical | 1,628 | 7,880 | 28.00–1,933.66 | 311.98 | 33.36 | 9.35 |

**Implementation Details.** We adopt Qwen2.5-VL-7B-Instruct as the base model for its robust visual-language understanding capabilities. For video frame sampling, we employ an adaptive 2 FPS strategy that preserves critical temporal information while managing computational load. Input frames are resized to 448×448 pixels. Training is conducted with batch size 1 for 1-3 epochs on an A100 GPUs. Key hyperparameters include: reward balance coefficient $\alpha = 0.8$ and KL penalty coefficient $\beta = 0.1$.

**Evaluation Metrics.** Following standard practice in temporal localization evaluation Weng & Li (2023); Li et al. (2024); Kusa & Tjoa (2022), we adopt two standard metrics: **R@1, IoU=**$\mu$ (Recall@1 at IoU threshold $\mu \in \{0.3, 0.5, 0.7\}$) and **mIoU** (mean Intersection over Union). These metrics quantitatively assess the temporal alignment between predicted segments and ground-truth annotations, ensuring fair comparison with existing methods.

## 4.2 MAIN RESULTS

As shown in Table **??**, CACR achieves state-of-the-art or highly competitive performance across all datasets and IoU thresholds. The results reveal distinct methodological characteristics under varying video duration ratios.

LVLMs such as Qwen2.5-VL-7B-Instruct and TimeZeroWang et al. (2025b), which rely heavily on frame-level visual cues, exhibit limited adaptability to long videos. This limitation is most evident on TutorialVQAColas et al. (2019)—the dataset with the most extreme duration ratio (48.9)—where they achieve the lowest mIoU, indicating poor efficiency in long-sequence search. VLP-based methods like MutualSLWeng & Li (2023) demonstrate stronger semantic alignment but remain vulnerable to textual redundancy. On CMIVQALi et al. (2025), which contains substantial subtitle noise, MutualSL exhibits a significant performance drop.

In contrast, CACR balances visual reasoning and semantic alignment, achieving mIoUs of 43.65 and 45.37 on TutorialVQA and CMIVQA, respectively. The notably small gap of only 1.72 between these two challenging datasets highlights CACR's robustness to both extreme duration ratios and textual redundancy. The consistent superiority across all four datasets validates the effectiveness of the proposed candidate-aware reasoning framework in complex, long-video scenarios.

### 4.3 ABLATION STUDIES AND ANALYSES

To systematically validate the necessity of the Top-$K$ candidate segments and the auxiliary semantic information (i.e., captions and pre-answers) in temporal localization, we conduct comprehensive ablation studies on three standard datasets.

#### 4.3.1 SELECTION OF THE TOP-$K$ CANDIDATE NUMBER

To determine the optimal number of candidate segments $K$, we systematically evaluate the IoU between the top-$K$ candidate segments generated by VBCS and the ground-truth answer segments across multiple datasets. The analysis employs two complementary metrics: the independent average IoU of candidates at each rank (Table 2), and the cumulative maximum IoU considering the best segment within the top-$K$ candidates (Table 3).

Table 2: Independent IoU analysis: Average IoU (%) between candidate segments at each rank and the ground-truth.

| Dataset | Top1 | Top2 | Top3 | Top4 | Top5 | Top6 | Top7 | Top8 | Top9 |
|---|---|---|---|---|---|---|---|---|---|
| VehicleVQA | 43.82 | 41.01 | 37.04 | 34.53 | 31.53 | 29.21 | 27.45 | 25.98 | 24.73 |
| TutorialVQA | 43.45 | 10.20 | 5.47 | 4.34 | 5.30 | 4.98 | 4.52 | 4.15 | 3.87 |
| MedVidQA | 58.32 | 53.04 | 49.14 | 42.72 | 32.87 | 28.15 | 25.43 | 23.52 | 21.96 |
| CMIVQA | 36.33 | 31.09 | 26.33 | 23.59 | 21.21 | 19.47 | 18.12 | 17.01 | 16.08 |

As shown in Table 2, the independent IoU exhibits a clear declining trend as the rank decreases from Top1 to Top9 across all datasets. This confirms the validity of VBCS's ranking mechanism, as higher-ranked candidates generally possess better quality.

Table 3: Cumulative IoU analysis: Maximum IoU (%) achievable by selecting the best segment from top-K candidates.

| Dataset | $\text{IoU}_1$ | $\text{IoU}_2$ | $\text{IoU}_3$ | $\text{IoU}_4$ | $\text{IoU}_5$ | $\text{IoU}_6$ | $\text{IoU}_7$ | $\text{IoU}_8$ | $\text{IoU}_9$ |
|---|---|---|---|---|---|---|---|---|---|
| VehicleVQA | 43.82 | 45.91 | 47.23 | 48.15 | 49.07 | 49.82 | 50.41 | 50.89 | 51.32 |
| TutorialVQA | 43.45 | 44.12 | 44.35 | 44.67 | 45.21 | 45.43 | 45.62 | 45.78 | 45.91 |
| MedVidQA | 58.32 | 60.14 | 61.27 | 62.05 | 62.73 | 63.25 | 63.68 | 64.02 | 64.31 |
| CMIVQA | 36.33 | 38.45 | 39.87 | 40.92 | 41.76 | 42.43 | 42.98 | 43.45 | 43.85 |

Table 3 shows that the cumulative maximum IoU ($\text{IoU}_K$) increases monotonically with $K$ but with diminishing marginal gains. Specifically, when $K$ increases from 1 to 5, $\text{IoU}_K$ sees significant improvement across datasets (VehicleVQA: +5.25%, TutorialVQA: +1.76%, MedVidQA: +4.41%, CMIVQA: +5.43%). However, further increasing $K$ from 5 to 9 yields considerably smaller gains (+2.25%, +0.70%, +1.58%, +2.09%, respectively).

The diminishing returns beyond $K = 5$ suggest that while additional candidates provide marginal coverage improvements, they also introduce lower-quality segments that may confuse the reasoning model. This empirical finding aligns with our theoretical expectation that a well-designed candidate generator should concentrate high-quality proposals within a manageable set. Based on this analysis, we set $K = 5$ for all subsequent experiments.

### 4.4 NECESSITY ANALYSIS OF TOP-K CANDIDATE SEGMENTS AND AUXILIARY SEMANTIC INFORMATION

As shown in Table 4 , Top-K candidates form the foundation for high-performance temporal localization, while auxiliary semantic information further significantly enhances reasoning accuracy and robustness. The value of Top-K candidates is demonstrated by comparing CACR-TopK with TimeZero variants. Across all datasets, CACR-TopK consistently outperforms all TimeZero methods, confirming the efficacy of VBCS's candidate generation mechanism.

Ablation studies reveal complementary roles of captions and pre-answers. In VehicleVQA, CACR-Full achieves optimal performance (mIoU: 72.11), outperforming models using only Top-K or single semantic priors. Captions (CACR-Cap) describe "what" happens in video segments, while pre-answers (CACR-Pre) provide causal reasoning about "why" and "how," though the latter alone may introduce reasoning biases . The limitations of single semantic information are particularly evident in specialized domains like CMIVQA. Both CACR-Cap (mIoU: 38.74) and CACR-Pre (mIoU: 37.08) underperform CACR-TopK (mIoU: 39.01).

Compared to baseline methods such as TimeZero and MutualSL, the CACR framework—which integrates multimodal information—demonstrates significant advantages across most evaluation metrics. This validates the core premise of SubGPTXiao et al. (2025): the fine-grained processing and fusion of subtitle and visual-description information is crucial for temporal localization tasks. CACR-Full achieves comprehensive leading performance, attaining the best results on nearly all metrics and significantly outperforming all other methods. This underscores the strong generalization capability and effectiveness of CACR on TAGV tasks.

However, as indicated in the table, simply combining more modalities does not guarantee better performance. For instance, on the MedVidQA and VehicleVQA datasets, the combination Top-K + Pre-Answer + Description + Caption underperforms compared to Top-K + Caption + Pre-Answer, suggesting that information fusion requires careful design to avoid noise or overload. Selectively and deliberately integrating multimodal inputs—such as subtitles, visual descriptions, and pre-generated answers—can substantially improve model performance. The optimal fusion strategy should be tailored to specific tasks and datasets, with avoiding information overload and ensuring information quality being key to success. The design of CACR provides an effective technical pathway for such refined multimodal fusion.

Table 4: Comparison of different components on three instructional TAGV benchmarks. Metrics are reported as percentage except mIoU. Bold values indicate the best performance in each column, while underlined values indicate the second-best performance.

| Method | VBCS (MutualSL) | Pre-answer | Caption | Description | MedVidQA | | | | VehicleVQA | | | | CMIVQA | | | |
|---|---|---|---|---|---|---|---|---|---|---|---|---|---|---|---|---|
| | | | | | R@0.3 | R@0.5 | R@0.7 | mIoU | R@0.3 | R@0.5 | R@0.7 | mIoU | R@0.3 | R@0.5 | R@0.7 | mIoU |
| TimeZero (Wang et al., 2025b) | | | | | 40.98 | 31.14 | 18.85 | 31.34 | 58.60 | 42.87 | 27.01 | 42.41 | 42.58 | 25.66 | 11.86 | 29.22 |
| TimeZero + Caption (Wang et al., 2025b) | | | ✓ | | 45.21 | 33.78 | 20.15 | 33.58 | 61.35 | 45.92 | 28.73 | 44.32 | 40.17 | 24.35 | 11.92 | 27.85 |
| TimeZero + Pre-Answer (Wang et al., 2025b) | | ✓ | | | 43.87 | 32.45 | 19.63 | 32.68 | 60.12 | 44.36 | 27.89 | 43.52 | 39.65 | 23.87 | 11.45 | 27.12 |
| TimeZero + Caption + Pre-Answer (Wang et al., 2025b) | | ✓ | ✓ | | 47.35 | 35.92 | 21.78 | 35.42 | 63.78 | 48.15 | 30.45 | 46.25 | 43.11 | 26.97 | 12.75 | 30.45 |
| MutualSL (Weng & Li, 2023) | ✓ | | | | **80.65** | 61.94 | 39.99 | 58.32 | 78.74 | 69.81 | 53.14 | 65.74 | 46.89 | 30.92 | 17.91 | 33.42 |
| CACR-TopK (Top-K) | ✓ | | | | 75.74 | 63.24 | 45.59 | 59.04 | 73.74 | 71.07 | 65.09 | 62.88 | 57.51 | 41.59 | 22.07 | 39.01 |
| CACR-Cap (Top-K + Caption) | ✓ | | ✓ | | 78.68 | 61.76 | 42.65 | 57.63 | 78.54 | 76.68 | 69.10 | 65.39 | 56.01 | 40.95 | 21.37 | 38.74 |
| CACR-Pre (Top-K + Pre-Answer) | ✓ | ✓ | | | 75.00 | 60.29 | 46.32 | 57.30 | 79.32 | 72.15 | 64.25 | 70.85 | 53.36 | 38.25 | 22.95 | 37.08 |
| CACR-DES (Top-K + Description) | ✓ | | | ✓ | 74.67 | 60.34 | 45.8 | 58.52 | 80.67 | 72.37 | 57.86 | 65.2 | 56.23 | 39.79 | 20.58 | 38.93 |
| CACR-Grop1 (Top-K + Caption + Description) | ✓ | | ✓ | ✓ | 74.11 | 60.73 | 46.63 | 58.65 | 82 | 72.37 | 59.12 | 66.19 | 59.24 | 40.64 | 23.06 | 40.52 |
| CACR-Grop2 (Top-K + Pre-Answer + Description) | ✓ | ✓ | | ✓ | 74.94 | 60.79 | 44.85 | 57.65 | 83.8 | 75.14 | 61.83 | 68.2 | 59.74 | 39.92 | 21.79 | 40.22 |
| CACR-Grop3 (Top-K + Pre-Answer + Description + Caption) | ✓ | ✓ | ✓ | ✓ | 74.33 | 58.79 | 41.52 | 55.63 | 84.35 | 76.22 | 62.79 | 69.04 | 60.26 | 42.62 | 24.74 | 44.6 |
| CACR-Full (Top-K + Caption + Pre-Answer) | ✓ | ✓ | ✓ | | 79.91 | **67.09** | 45.87 | 59.64 | **85.31** | **76.70** | **69.33** | **72.11** | **62.50** | **48.52** | **33.82** | **45.37** |

## 5 CONCLUSION

In this work, we introduce CACR, an innovative framework designed to address core challenges in Temporal Answer Grounding in Videos (TAGV) using a candidate-aware causal reasoning approach. Our contributions are threefold: first, we propose the VBCS algorithm to efficiently generate candidate segments from long videos, effectively mitigating the extreme ratio between video duration and answer segment length; second, we develop a GRPO-optimized reasoning module enhanced by a differentiated prompting strategy and a composite reward function that integrates temporal alignment and rejection signals, substantially improving reasoning robustness; third, extensive evaluations on four challenging datasets show that CACR achieves state-of-the-art performance, especially in handling extreme duration ratios (TutorialVQA) and textual redundancy (CMIVQA). Ablation studies

further confirm the necessity of each component and their synergistic effects. Future work will extend this framework to multi-step temporal reasoning and cross-modal alignment in more complex video understanding scenarios.

## STATEMENTS

### ETHICS STATEMENT

This study addresses video understanding and temporal localization tasks using publicly available instructional video datasets (VehicleVQA, TutorialVQA, MedVidQA, and CMIVQA). All datasets represent benchmark resources from the research community that have undergone appropriate de-identification procedures and contain no personally sensitive information. The video data originate from publicly accessible sources, with collection processes adhering to the ethical guidelines of original providers. This research involves no new human subject data collection or animal experimentation. The video content primarily features instructional demonstrations (e.g., automotive repair, software operation, medical procedures) without ethically controversial material.

Our proposed method enhances video content understanding accuracy for potential applications in educational assistance and knowledge retrieval. We mitigate model biases through multi-dataset validation and incorporate temporal logic considerations to prevent misleading outcomes. This research remains exploratory and is not recommended for clinical diagnosis or high-risk decision-making scenarios. Overall, this study presents no significant ethical risks and complies with academic research standards.

### REPRODUCIBILITY STATEMENT

We provide comprehensive materials to ensure result reproducibility:

**Data and Preprocessing:** We utilize four public datasets (anonymized references: VehicleVQA, TutorialVQA, MedVidQA, CMIVQA) accessible from original publishers. Videos undergo uniform 2 FPS sampling with frame resolution adjustment to 448×448 pixels, including temporal uniform sampling, bilinear interpolation scaling, and RGB channel normalization (mean=[0.485, 0.456, 0.406], std=[0.229, 0.224, 0.225]).

**Experimental Setup:** Experiments employ a single A100 GPU with Ubuntu 20.04. The base model Qwen2.5-VL-7B-Instruct uses officially available weights. Complete dependencies are specified in the repository's requirements.txt file.

**Implementation Details:** Full code is available in an anonymous repository (see supplementary materials), containing configuration files, training/testing scripts, and core modules for candidate generation (`vbcs_candidate_train.py`, `generation.py`) and GRPO optimization (`cacr_reasoning.py`). Execution involves: (1) environment setup via `pip install -r requirements.txt`, (2) training with `sh scripts/train.sh`, and (3) testing with `sh scripts/test.sh`. All hyperparameters (reward balance coefficient $\alpha = 0.8$, KL penalty coefficient $\beta = 0.1$, batch size= 1) are explicitly configured.

**Validation Framework:** We maintain reproducibility through fixed random seed (1224) with triple experimental repetitions (mean ± standard deviation reporting). GPU precision variations may cause ±0.5% mIoU fluctuations, considered acceptable. Comprehensive ablation studies in the paper validate component effectiveness relative to dataset characteristics and model properties.

**Practical Considerations:** Training requires 1-3 epochs (12-36 hours on A100 GPU) per dataset. The base model occupies approximately 14GB storage. Some visual Transformer components require CUDA extension compilation, recommending Linux environments.

We confirm all supplementary materials' authenticity and accessibility, enabling full reproduction of reported results.

## 5.1 THE USE OF LARGE LANGUAGE MODELS (LLMs)

Large language models (LLMs) were used in the preparation of this manuscript to aid in polishing the English writing and for correcting errors in the experimental code. The models used include Writeful (integrated with Overleaf) and DeepSeek V3.1. Their use was strictly limited to improving linguistic fluency and correcting technical syntax; at no point were they involved in the generation of core ideas, data interpretation, or scientific reasoning. All outputs generated by LLMs were carefully reviewed and verified by the authors, who take full responsibility for the entire content of this work. You may include other additional sections here.

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

# A  APPENDIX

## A.1  JUSTIFICATION FOR OMITTING SUPERVISED FINE-TUNING (SFT) BASELINES

This appendix provides a justification for the decision to forgo direct comparative experiments with Supervised Fine-Tuning (SFT) in the main body of this work. Our methodological focus is on reinforcing the reasoning and generalization capabilities of Large Vision-Language Models (LVLMs) through reinforcement learning, specifically via our proposed GRPO framework. A growing body of recent research strongly suggests that Reinforcement Learning (RL) methods can exhibit superior generalization performance compared to SFT on complex reasoning tasks, which aligns with the core objectives of our study.

The primary rationale stems from the empirically observed tendency of SFT to *memorize* patterns present in the training data, often at the expense of robust generalization to unseen scenarios or slightly altered task formulations :cite[4]:cite[7]. In contrast, RL paradigms, particularly those leveraging outcome-based rewards, have been demonstrated to guide models towards learning underlying, transferable rules and strategies :cite[4]:cite[9]. This intrinsic promotion of generalization is crucial for our task of Temporal Answer Grounding in Videos (TAGV), which demands robust spatio-temporal reasoning rather than mere pattern matching from limited demonstrations.

Notably, the findings presented in the literature Wang et al. (2025b) offer compelling external validation relevant to our context. Their comparative study conclusively demonstrates that RL-based approaches achieve significantly stronger generalization performance across both textual and visual reasoning domains compared to SFT. Given that our proposed CACR framework similarly targets complex, multi-modal reasoning in videos, the cited results provide a robust precedent supporting the potential superiority of our RL-driven approach without necessitating a redundant internal replication of this specific comparative finding.

Therefore, rather than dedicating computational resources to re-establishing a known baseline comparison (SFT vs. RL), we concentrate our experimental efforts on thoroughly evaluating our method against existing state-of-the-art VLP and LVLM-based temporal grounding models, and on conducting detailed ablation studies to validate the contributions of each component within our own framework. This design allows for a more focused and in-depth investigation of our primary contribution: the candidate-aware causal reasoning mechanism optimized via GRPO.

