# OpenReview forum: "CACR: Reinforcing Temporal Answer Grounding in Videos via Candidate-Aware  Causal Reasoning"
_ICLR.cc/2026/Conference — Submitted to ICLR 2026_

### Official Review · Reviewer_zMGK · 2025-10-17

**Soundness:** 3
**Presentation:** 2
**Contribution:** 3
**Rating:** 6
**Confidence:** 3

**Summary:**

This paper addresses the task of Temporal Answer Grounding in Videos (TAGV), which involves identifying the precise video segment that answers a natural language query. To handle challenges such as complex question semantics and the large temporal mismatch between lengthy videos and short answer segments, the authors propose a Candidate-Aware Causal Reasoning (CACR) framework. The approach first utilizes a vision-language pre-trained (VLP) model to efficiently generate K candidate segments, followed by a temporal logic reasoning module enhanced with a rejection reward mechanism to refine segment selection. The entire model is optimized using Group Relative Policy Optimization (GRPO). Experimental results demonstrate that CACR achieves state-of-the-art performance in terms of mean Intersection over Union (mIoU) across four benchmarks for reasoning-based retrieval in long videos.

**Strengths:**

1. The paper is clearly written and well-organized.
2. The paper introduces a novel CACR framework that integrates GRPO to refine top-k candidate segments through deep multi-modal reasoning, offering a principled solution to the challenges of temporal grounding.
3. The proposed method demonstrates consistent and notable performance improvements across multiple TAGV benchmarks, indicating its effectiveness.

**Weaknesses:**

1. The paper suffers from a substantial number of typographical and formatting errors that detract from its readability and professionalism. For instance, several sentences lack proper spacing (e.g., L105, L273, L443), and the acronym GRPO is inconsistently defined as Generalized Relative Policy Optimization instead of Group Relative Policy Optimization (e.g., L24, L102). In addition, the paper omits parentheses around citations and is missing several citations, which should be carefully corrected in revision.
2. The proposed two-stage pipeline, comprising candidate generation followed by multi-modal reasoning with a large vision-language model, may introduce significant computational overhead relative to existing TAGV baselines. A detailed complexity analysis or efficiency comparison would strengthen the empirical justification for the approach.
3. The rationale for adopting pre-answer instead of caption-based descriptions (L270–L280) is insufficiently explained. Clarifying this design choice, possibly with an in-depth analysis, would support their claims.

**Questions:**

1. Does the model append the full video clip at each reasoning turn during the iterative process? If so, could you provide details regarding the number of visual tokens involved and how this affects computational efficiency and memory usage?

---

> ### Author Response · Authors · 2025-12-04
> **Response to Weakness 1: Corrections to Typographical, Formatting, and Citation Errors**
>
> We sincerely appreciate the feedback on improving the paper’s readability and professionalism. We fully acknowledge the issues of typographical errors, inconsistent formatting, and citation omissions, and have conducted a comprehensive review and correction of the entire manuscript.

---

> ### Author Response · Authors · 2025-12-04
> **Response to Weakness 2: Computational Complexity Analysis & Efficiency Comparison of the Two-Stage Pipeline**
>
> In response to reviewers’ concerns regarding the computational overhead of two-stage methods, we have conducted a detailed efficiency analysis across multiple datasets. The results demonstrate that our proposed CACR method consistently achieves a better balance between computational cost and localization accuracy when processing videos of varying lengths and characteristics, thanks to its two-stage design of “lightweight global candidate generation + targeted local deep reasoning.” Compared to large-scale end-to-end models that perform full-video inference (e.g., TimeZero), CACR achieves significant performance improvements with comparable or even lower total time cost.
>
> Below is a performance comparison of CACR, TimeZero, MutualSL, and VPTSL across six benchmarks:
>
> **Table D.1 A Performance Comparison of CACR, TimeZero, MutualSL, and VPTSL Across Six Benchmarks**
>
> | Dataset    | Method           | Avg. Processed Content (Duration)   | Total Inference Time (s) | Peak GPU Memory (MB) | R@0.3 | R@0.5 | R@0.7 | mIoU |
> | :--------- | :--------------- | :---------------------------------- | :----------------------- | :------------------- | :---- | :---- | :---- | :--- |
> | CMIVQA     | TimeZero         | Full Video (310s)                   | 4.35                     | 19,052               | 42.58 | 25.66 | 11.86 | 29.22 |
> |            | MutualSL         | Full Video (310s)                   | 0.11                     | 4700                 | 46.89 | 30.92 | 17.91 | 33.42 |
> |            | **CACR (Ours)**  | **Stage 1: 310s + Stage 2: ~32s**   | **3.037**                | **17737.2**          | **62.5** | **48.52** | **33.82** | **45.37** |
> | MedVidQA   | TimeZero         | Full Video (474s)                   | 4.613                    | 19652.1              | 40.98 | 31.14 | 18.85 | 31.34 |
> |            | MutualSL         | Full Video (474s)                   | 0.21                     | 4832                 | 80.65 | 61.94 | 39.99 | 58.32 |
> |            | **CACR (Ours)**  | **Stage 1: 474s + Stage 2: ~67s**   | **3.31**                 | **18289.2**          | **79.91** | **67.09** | **45.87** | **59.65** |
> | VehicleVQA | TimeZero         | Full Video (125s)                   | 3.37                     | 18203.2              | 58.6  | 42.87 | 27.01 | 42.41 |
> |            | MutualSL         | Full Video (125s)                   | 0.06                     | 4420.4               | 78.74 | 69.81 | 53.14 | 65.74 |
> |            | **CACR (Ours)**  | **Stage 1: 125s + Stage 2: ~22s**   | **2.86**                 | **17737.2**          | **85.31** | **76.7** | **69.33** | **72.11** |
> | TutorialVQA| TimeZero         | Full Video (287s)                   | 4.004                    | 18914.6              | 22.9  | 10.01 | 4.35  | 18.44 |
> |            | MutualSL         | Full Video (287s)                   | 0.13                     | 4676                 | 60.14 | 43.59 | 28.28 | 43.48 |
> |            | **CACR (Ours)**  | **Stage 1: 287s + Stage 2: ~6s**    | **2.07**                 | **17185.8**          | **60.31** | **43.65** | **28.72** | **43.65** |
> | COIN       | TimeZero         | Full Video (146s)                   | 3.124                    | 17943.7              | 29.26 | 15.89 | 5.36  | 20.05 |
> |            | VPTSL            | Full Video (146s)                   | 0.07                     | 4466                 | 30.13 | 18.67 | 9.39  | 22.09 |
> |            | **CACR (Ours)**  | **Stage 1: 146s + Stage 2: ~36s**   | **2.997**                | **17385.6**          | **59.11** | **43.81** | **24.88** | **41.06** |
> | Crosstalk  | TimeZero         | Full Video (298s)                   | 4.075                    | 18980.9              | 27.15 | 14.75 | 4.46  | 18.57 |
> |            | VPTSL            | Full Video (298s)                   | 0.13                     | 4682                 | 42.94 | 36.60 | 25.48 | 31.69 |
> |            | **CACR (Ours)**  | **Stage 1: 298s + Stage 2: ~10s**   | **2.077**                | **1807.8**           | **55.89** | **41.77** | **22.4** | **40.19** |
>
> In terms of time efficiency, CACR employs a “focused” reasoning strategy, resulting in significantly lower total inference time than TimeZero, which requires deep end-to-end processing of the entire video. For instance, on a 474-second video, CACR takes 3.31 seconds, saving approximately 28.3% compared to TimeZero’s 4.613 seconds. Although CACR introduces an additional 2–3 seconds of deep reasoning overhead compared to faster methods like MutualSL, this cost is justified by a substantial leap in performance. On the VehicleVQA dataset, for example, CACR achieves 76.7% R@0.5 in just 2.86 seconds, outperforming both MutualSL (69.81%) and TimeZero (42.87%) and representing a Pareto improvement in efficiency and accuracy.

---

> > ### Author Response · Authors · 2025-12-04
> > **Response to Weakness 2: Computational Complexity Analysis & Efficiency Comparison of the Two-Stage Pipeline**
> >
> > Regarding memory consumption, CACR generally exhibits lower peak GPU memory usage than TimeZero. This is because its first-stage model is extremely lightweight (about 1/40 the memory footprint of the second stage), and the second stage only processes short video segments. For example, on the Crosstalk dataset, CACR’s peak memory is 18,007.8 MB, lower than TimeZero’s 18,980.9 MB. By confining computationally intensive “heavy” processing to brief key segments, CACR avoids maintaining high memory states over extended periods, alleviates GPU memory bottlenecks, and results in smoother, more predictable memory usage—enhancing deployment feasibility and system stability.
> >
> > Importantly, these efficiency gains do not come at the expense of performance; instead, they lead to comprehensive improvements. Across all six datasets and under every evaluation metric (R@0.3, R@0.5, R@0.7, mIoU), CACR consistently and substantially outperforms TimeZero. Moreover, except for matching fast baselines on TutorialVQA, CACR surpasses methods like MutualSL in key accuracy metrics on the remaining five datasets. This confirms the necessity of the second-stage deep reasoning, whose computational overhead is efficiently and directly translated into tangible gains in localization precision.
> >
> > **In summary**, CACR does not merely introduce redundant computation; rather, it represents an intelligent computing paradigm tailored to the characteristics of long video localization. By employing an efficient lightweight front-end for rapid global scanning and accurately guiding deep models to focus on the most relevant local segments, CACR achieves comprehensive improvements over existing single-stage methods in inference speed, memory footprint, and final accuracy. Its design is well-justified and empirically validated, offering an effective pathway for co-optimizing efficiency and precision in this field.

---

> ### Author Response · Authors · 2025-12-04
> **Response to Weakness 3: Rationale for Adopting Pre-answer Over Caption-Based Descriptions**
>
> **1. Experimental Validation**
> We have conducted ablation experiments to analyze the impact of three key modal inputs—captions, visual descriptions, and pre-answers—on the CACR framework. The comparative results are summarized in Table 5 (included in the manuscript), which clearly demonstrates the contribution of each component. Detailed analyses are provided in the method and experiment sections.
>
> **Table 5: Comparison of different components on three TAGV benchmarks.**
>
>
> | Method | VBCS(MutualSL) | Pre-answer | Caption | Description | MedVidQA | | | | VehicleVQA | | | | CMIVQA | | | |
> | :--- | :---: | :---: | :---: | :---: | :---: | :---: | :---: | :---: | :---: | :---: | :---: | :---: | :---: | :---: | :---: | :---: |
> | | | | | | R@0.3 | R@0.5 | R@0.7 | mIoU | R@0.3 | R@0.5 | R@0.7 | mIoU | R@0.3 | R@0.5 | R@0.7 | mIoU |
> | TimeZero(Wang et al.2025b) | | | | | 40.98 | 31.14 | 18.85 | 31.34 | 58.60 | 42.87 | 27.01 | 42.41 | 42.58 | 25.66 | 11.86 | 29.22 |
> | TimeZero + Caption (Wang et al.2025b) | | | ✓ | | 45.21 | 33.78 | 20.15 | 33.58 | 61.35 | 45.92 | 28.73 | 44.32 | 40.17 | 24.35 | 11.92 | 27.85 |
> | TimeZero + Pre-Answer (Wang et al.. 2025b) | | | | | 43.87 | 32.45 | 19.63 | 32.68 | 60.12 | 44.36 | 27.89 | 43.52 | 39.65 | 23.87 | 11.45 | 27.12 |
> | TimeZero + Caption + Pre-Answer (Wang et al.,2025b) | | ✓ | ✓ | | 47.35 | 35.92 | 21.78 | 35.42 | 63.78 | 48.15 | 30.45 | 46.25 | 43.11 | 26.97 | 12.75 | 30.45 |
> | MutualSL(Weng & Li, 2023) | | | | | 80.65 | 61.94 | 39.99 | 58.32 | 78.74 | 69.81 | 53.14 | 65.74 | 46.89 | 30.92 | 17.91 | 33.42 |
> | CACR-TopK(Top-K) | ✓ | | | | 75.74 | 63.24 | 45.59 | 59.04 | 73.74 | 71.07 | 65.09 | 62.88 | 57.51 | 41.59 | 22.07 | 39.01 |
> | CACR-Cap(Top-K+Caption) | ✓ | | | | 78.68 | 61.76 | 42.65 | 57.63 | 78.54 | 76.68 | 69.10 | 65.39 | 56.01 | 40.95 | 21.37 | 38.74 |
> | CACR-Pre (Top-K +Pre-Answer) | ✓ | | | | 75.00 | 60.29 | 46.32 | 57.30 | 79.32 | 72.15 | 64.25 | 70.85 | 53.36 | 38.25 | 22.95 | 37.08 |
> | CACR-DES(Top-K+Description) | ✓ | | | ✓ | 74.67 | 60.34 | 45.8 | 58.52 | 80.67 | 72.37 | 57.86 | 65.2 | 56.23 | 39.79 | 20.58 | 38.93 |
> | CACR-Grop1(Top-K+Caption+Description) | ✓ | | ✓ | ✓ | 74.11 | 60.73 | 46.63 | 58.65 | 82 | 72.37 | 59.12 | 66.19 | 59.24 | 40.64 | 23.06 | 40.52 |
> | CACR-Grop2(Top-K+Pre-Answer+Description) | ✓ | ✓ | | ✓ | 74.94 | 60.79 | 44.85 | 57.65 | 83.8 | 75.14 | 61.83 | 68.2 | 59.74 | 39.92 | 21.79 | 40.22 |
> | CACR-Grop3(Top-K+Pre-Answer+Description+Caption) | ✓ | ✓ | ✓ | ✓ | 74.33 | 58.79 | 41.52 | 55.63 | 84.35 | 76.22 | 62.79 | 69.04 | 60.26 | 42.62 | 24.74 | 44.6 |
> | **CACR-Full(Top-K+Caption+Pre-Answer)** | **✓** | **✓** | **✓** | | **79.91** | **67.09** | **45.87** | **59.64** | **85.31** | **76.70** | **69.33** | **72.11** | **62.50** | **48.52** | **33.82** | **45.37** |
>
> **2. Rationale for Using Pre-answers Rather Than Caption-based Descriptions (L270–L280)**
> The decision to employ pre-answers instead of caption-based descriptions is grounded in the inherent challenges of the TAGV task and the need to support effective supervised learning. Below we elaborate on three key aspects:
>
> **(1) Core Task Challenge: Complexity of TAGV and Representation Requirements in Supervised Learning**
> TAGV requires reasoning from an abstract question to a temporal answer in video content—a process that involves multi-step inference. Under supervised training, models must learn a mapping from question understanding → answer hypothesis generation → video evidence verification. Relying only on raw video features and question text fails to build a semantic bridge between abstract queries and concrete visual content, limiting learning efficiency and generalization.
>
> **(2) Limitation of Single Information Sources in Supervised Training**
> Training with incomplete inputs leads to biased learning and weak reasoning:
> - **Visual-only inputs** (e.g., visual descriptions) provide “what is in the video” but lack explicit guidance. Models may learn superficial visual matching rather than causal reasoning.
> - **Text-only inputs** (e.g., questions) risk causing the model to rely purely on textual logic while ignoring alignment with actual video evidence.
> Thus, effective supervised training requires both a reliable, structured account of video content and a clear, target-oriented verification signal.

---

> ### Author Response · Authors · 2025-12-04
> **Response to Weakness 3: Rationale for Adopting Pre-answer Over Caption-Based Descriptions**
>
> **(3) Necessity of Combined Inputs in CACR-Full**
> Our CACR‑Full design (Top‑K + caption-based subtitle + pre-answer) creates a layered semantic input that optimizes supervised learning:
> - **Caption-based semantics** offer an objective, timeline-aligned summary of video content. They reduce the representational burden of parsing raw multimodal data, allowing the model to focus on learning the reasoning mapping.
> - **Pre-answers** act as a **target-driven prior**, generated by an LLM from the question. They provide a hypothetical description of the answer, guiding the model to verify “what should be looked for” rather than merely “what is seen.”
> - **Synergy** creates a “hypothesis–observation” alignment: pre-answers define the target, captions provide grounded observations, and the policy network learns to align them under the supervision of ground-truth temporal boundaries. The reference model, trained only on pre-answers, maintains a goal-driven reasoning pattern regularized via KL divergence, ensuring the policy model does not degenerate into superficial matching.
>
> In summary, the choice of pre-answers alongside caption-based semantics is deliberate: it furnishes the model with both **goal-oriented hypotheses** and **observation-grounded facts**, constructing a rich, hierarchical input representation. This enables CACR to learn the “hypothesis–observation–verification” reasoning more effectively under supervision, leading to more accurate and robust temporal answer grounding. The design directly addresses the semantic gap in TAGV and strengthens generalization by shaping the learning process toward causal reasoning.

---

> ### Author Response · Authors · 2025-12-04
> **Response to Question1:Information on Video Sampling and Visual Token Count in Reasoning Turns**
>
> The model does not append the full video clip at each reasoning turn. Instead, it employs a two-stage strategy where a lightweight global scan first identifies key segments, and then only a single, short candidate clip is processed in detail during each subsequent reasoning step. This approach avoids the computational burden of repeatedly processing the entire video.
>
> **Stage 1: Coarse-grained Global Scan (VBCS Module)**
> The VBCS module aims to quickly understand videos of arbitrary length at a constant computational cost. Its visual token generation is independent of spatial resolution and depends only on duration, achieving a sparse representation of “one token per second.”
>
> *Token Generation Mechanism*
> Given a video with total duration $L_\text{total}$ (seconds), this module uses a pretrained I3D network to extract features with a fixed temporal window. The total number of visual tokens $T_\text{vbcs}$ is determined by:
> $T_\text{vbcs} = \lceil (L_\text{total} \times f_\text{sampling}) / w_\text{size} \rceil$
> where $f_\text{sampling} = 16$ fps is the fixed sampling rate, and $w_\text{size} = 16$ frames is the I3D sliding window size. Each output token $v_i \in \mathbb{R}^{1024}$ encodes a 1-second spatiotemporal summary of the video. For example, a 30-minute (1800-second) video yields $T_\text{vbcs} = 1800$ tokens, forming a feature matrix $V_\text{vbcs} \in \mathbb{R}^{1800 \times 1024}$.
>
> *Design Advantages*
> This fixed sparse strategy ensures predictable overhead and very low linear growth ($O(L_\text{total})$). It enables efficient global semantic encoding of very long videos and quickly screens the Top-K (e.g., K=5) most relevant candidate clips, providing precise attention foci for the second stage.
>
> *Impact on Efficiency & Memory*
> - **Computationally efficient**: Token count grows linearly with duration and is independent of spatial resolution, enabling rapid scanning of videos lasting tens of minutes or even hours.
> - **Memory-friendly**: The total visual features to cache ($T\_\text{vbcs} \times 1024$) are minimal, exerting negligible memory pressure.

---

> > ### Author Response · Authors · 2025-12-04
> > **Response to Question1:Information on Video Sampling and Visual Token Count in Reasoning Turns**
> >
> > **Stage 2: Fine-grained Clip Reasoning (LVLM Module)**
> > The LVLM module conducts detailed analysis on each candidate clip. Unlike the first stage, the visual token count here is the result of dynamic optimization among spatial resolution, temporal sampling rate, and the model’s sequence length limit, aiming to allocate the densest possible tokens to key clips within a limited “token budget” to support fine-grained understanding.
> >
> > *Adaptive Tokenization Process*
> > For a candidate clip of length $L_\text{clip}$ seconds, tokenization follows these adaptive steps:
> > 1. **Temporal sampling**: Determine the base frame count $N\_\text{base} = L_\text{clip} \times \text{FPS}_\text{target}$ (e.g., 2 fps), round it to the nearest power of 2, and clamp it within $[4, 768]$ to obtain the final sampled frame count $N_\text{frames}$.
> > 2. **Compute pixel budget**: This is the core constraint step. Based on the sequence length limit, the model allocates a total pixel budget for all frames of the current clip and computes the **maximum pixels per frame**:
> > $\text{MaxPixelsPerFrame} \approx \min \left( \text{VIDEO\_FRAME\_MAX\_PIXELS}, \frac{0.9 \times \text{MODEL\_SEQ\_LEN} \times (\text{image\_factor})^2}{N_\text{frames} \times \text{FRAME\_FACTOR}} \right)$
> > where $\text{image\_factor}=28$. The parameter total_pixels (set to $3584 \times 28 \times 28$ in our system) defines the theoretical pixel upper bound based on sequence length.
> > 3. **Spatial resolution adjustment**: While preserving the aspect ratio, an intelligent scaling function adjusts the per-frame resolution to satisfy: (a) height and width are divisible by 28; (b) total pixels are between the defined lower bound (min_pixels = $16 \times 28 \times 28$) and the computed MaxPixelsPerFrame.
> > 4. **Generate visual tokens**: Based on the final resolution $(H, W)$ and the vision Transformer’s patch size (14), compute tokens per frame $\text{TokensPerFrame} = (H/14) \times (W/14)$. The total visual tokens for the clip are $T_\text{lvlm} = N_\text{frames} \times \text{TokensPerFrame}$, which must satisfy $T_\text{lvlm} \leq 0.9 \times \text{MODEL\_SEQ\_LEN}$.
> >
> > *Design Advantages*
> > The adaptive strategy of the LVLM module achieves an optimal trade-off between temporal and spatial information under a fixed sequence length budget. For longer clips, the system tends to moderately reduce per-frame resolution (decreasing $\text{TokensPerFrame}$) to accommodate more sampled frames (increasing $N_\text{frames}$), thereby preserving more complete temporal dynamics. This dynamic allocation ensures that precious model capacity is concentrated on the most informative visual content.
> >
> > *Resource Allocation*
> > - **Computation focused**: Avoids processing irrelevant frames. For instance, the model may allocate up to 70% of the sequence length budget to analyze a 20-second clip (generating thousands of tokens) rather than the entire long video, enabling unprecedented detail analysis.
> > - **Controlled memory usage**: Each iteration only loads and processes the visual tokens of one short clip, reducing memory consumption by orders of magnitude compared to processing the full video, making complex, multi-turn reasoning feasible with limited resources.
> >
> > **Synergistic Workflow and Efficiency Comparison**
> > The two-stage strategy, through division of labor and collaboration, implements an efficient pipeline from “finding a needle in a haystack” to “precision microscopy.” The following example illustrates the efficiency contrast: For a 30-minute (1800-second) video, the VBCS module uses only 1800 tokens (1 token/sec) to complete a global scan and locate 5 key clips. Subsequently, the LVLM module can allocate up to 7,680 visual tokens to a 30-second clip (achieved via 60 frames at 224×224 resolution), resulting in a token density (~256 tokens/sec) over 200 times higher than the first stage, thereby enabling deep understanding.
> >
> > **Conclusion**
> > The model does **not** append the full video clip at each reasoning turn. The VBCS module performs a one-time, low-density global scan (with a fixed, sparse set of tokens) for efficient candidate localization. Subsequently, during each iteration, the LVLM module only tokenizes a single candidate clip with high density and adaptation, thereby focusing core computational and memory resources precisely on the most valuable information segments under strict constraints. This synergistic design of “global coarse screening followed by local fine analysis” is key to the system’s ability to achieve both efficient processing and deep analysis in long-form video understanding tasks.

---

### Official Review · Reviewer_YQx5 · 2025-10-22

**Soundness:** 2
**Presentation:** 2
**Contribution:** 2
**Rating:** 4
**Confidence:** 4

**Summary:**

This paper introduces CACR, a framework for Temporal Answer Grounding in Videos (TAGV) that combines candidate segment generation with reinforcement learning-based reasoning. The authors address the challenging problem of locating short answer segments in long, untrimmed videos by proposing a two-stage approach: first, a VLP-based candidate selection (VBCS) module generates high-quality candidate segments, and second, a GRPO-optimized reasoning module performs fine-grained temporal localization using a composite reward function. The method is evaluated on four instructional video datasets and demonstrates state-of-the-art or competitive performance, particularly in handling extreme duration ratios and textual redundancy.

**Strengths:**

1)	The paper is built on a key observation: the maximum IoU between top-K candidates and ground truth increases with K. This motivates the use of a candidate-aware reasoning framework, which is both intuitive and empirically supported.

2)	The combination of VBCS for candidate generation and GRPO-based reasoning is well-justified. The use of a rejection reward mechanism and differentiated prompting strategy for the reference model are innovative and improve robustness.

3)	The method achieves SOTA or competitive results across four challenging datasets, with particularly impressive performance on TutorialVQA and CMIVQA, which involve extreme duration ratios and subtitle noise.

**Weaknesses:**

1)	While the paper compares with several VLP and LVLM methods, it would benefit from a more direct comparison with recent LVLM-based temporal grounding models (e.g., TimeScope: Towards Task-Oriented Temporal Grounding In Long Videos; Video-STR: Reinforcing MLLMs in Video Spatio-Temporal Reasoning with Relation Graph, etc) using the same candidate sets or reasoning mechanisms.

2)	The experiments are limited to instructional video datasets. It is unclear how well CACR generalizes to other domains (e.g., sports, movies, surveillance).

3)	The paper mentions using LLMs for captioning and pre-answer generation but does not specify which LLMs were used or their impact on performance. More details would improve reproducibility.

4)	While GRPO is motivated by prior success in reasoning tasks, a deeper theoretical or intuitive explanation of why it is particularly suitable for TAGV would be helpful.

5)	The details are hard to understand in Figure 1.

**Questions:**

See the weakness.

---

> ### Author Response · Authors · 2025-12-04
> **Response to Weakness 1: Lack of Recent LVLM-Based Temporal Grounding Models**
>
> Thank you for the reviewer's valuable suggestions. Your insight regarding conducting more direct comparisons with the latest LVLM-based temporal grounding models is highly insightful, as it helps to more clearly delineate the contributions of our work. We have carefully considered this point and have made the following refinements in the revised version:
>
> Our research focuses on the Temporal Answer Grounding in Videos (TAGV) task, where the input is a "question that requires visual demonstration," such as "How to change a bicycle tire?" This type of task typically corresponds to procedural, step-by-step instructional videos, requiring the model to locate the video segment that answers the question. In contrast, TimeScope addresses the Temporal Sentence Grounding in Videos (TSGV) task, where the input is a "phrase or sentence describing the visual content of the video," such as "A person runs first and then stretches." This task does not involve question-answering logic; its core is to locate the corresponding temporal interval in a long video based on natural language descriptions, emphasizing fine-grained alignment between semantics and visual content. Meanwhile, Video-STR focuses on video spatio-temporal reasoning, with the primary goal of answering deep-level questions about the spatial attributes, temporal dynamics, and their coupling relationships of objects in videos. The output is typically numerical values, coordinates, motion state descriptions, or multiple-choice answers, which is not directly applicable to the TAGV task we study.
>
> Following your suggestion, we have incorporated several state-of-the-art approaches—including SubGPT [1], TimeScope [2], Ask2Loc [3], TFVTG [4], VTG-R1 [5], and Time-R1 [6]—into our comparison. The updated results are presented in Table 2 (Comparison with prior methods on six TAGV benchmarks).

---

> > ### Author Response · Authors · 2025-12-04
> > **Response to Weakness 1: Lack of Recent LVLM-Based Temporal Grounding Models**
> >
> > **Table 2: Comparison with prior methods on six  TAGV benchmarks**
> >
> > | Family | Method | MedVidQA | | | | VehicleVQA | | | | CMIVQA | | | | TutorialVQA | | | | COIN | | | | CrossTask | | | |
> > | :--- | :--- | :---: | :---: | :---: | :---: | :---: | :---: | :---: | :---: | :---: | :---: | :---: | :---: | :---: | :---: | :---: | :---: | :---: | :---: | :---: | :---: | :---: | :---: | :---: | :---: |
> > | | | **R@0.3** | **R@0.5** | **R@0.7** | **mIoU** | **R@0.3** | **R@0.5** | **R@0.7** | **mIoU** | **R@0.3** | **R@0.5** | **R@0.7** | **mIoU** | **R@0.3** | **R@0.5** | **R@0.7** | **mIoU** | **R@0.3** | **R@0.5** | **R@0.7** | **mIoU** | **R@0.3** | **R@0.5** | **R@0.7** | **mIoU** |
> > | **Random** | Random Mode | 8.38 | 1.93 | 1.21 | 6.89 | 6.52 | 2.75 | 1.54 | 5.22 | 5.71 | 4.65 | 3.58 | 3.97 | 6.53 | 2.46 | 0 | 5.26 | 4.92 | 0.76 | 0.28 | 3.12 | 3.57 | 0.54 | 0.20 | 2.79 |
> > | | VSLBase (Zhang et al.,2021) | 27.66 | 14.19 | 6.99 | 21.01 | 18.95 | 8.64 | 4.28 | 20.11 | - | - | - | - | 10.84 | 9.58 | 0.37 | 8.71 | - | - | - | - | - | - | - | - |
> > | | VSLNet (Zhang et al.,2020a) | 30.32 | 16.61 | 8.39 | 22.41 | 16.53 | 8.47 | 4.03 | 20.07 | - | - | - | - | 11.03 | 9.93 | 0.66 | 9.58 | 13.11 | 7.99 | 4.36 | 9.06 | 19.04 | 12.76 | 8.32 | 13.22 |
> > | | TMLGA (Rodriguez-Opazo et al., 2020) | 23.87 | 14.84 | 6.21 | 20.49 | 17.69 | 8.79 | 3.46 | 16.54 | - | - | - | - | 10.39 | 9.24 | 0.34 | 8.65 | - | - | - | - | - | - | - | - |
> > | **VLP** | ACRM (Tang et al., 2021) | 24.83 | 16.55 | 10.96 | 22.89 | 20.77 | 12.1 | 8.27 | 22.28 | - | - | - | - | 12.61 | 11.37 | 1.26 | 11.12 | - | - | - | - | - | - | - | - |
> > | | MoR (Kusa & Tjoa, 2022) | 47.1 | 27.74 | 10.97 | 30.67 | 42.75 | 31.54 | 25.97 | 44.81 | - | - | - | - | 23.45 | 15.01 | 8.55 | 19.48 | 14.64 | 8.81 | 3.64 | 10.66 | 23.66 | 18.83 | 10.96 | 16.92 |
> > | | VTPSL (Li et al., 2024a) | 77.42 | 61.94 | 44.52 | 57.81 | 74.15 | 67.15 | 54.59 | 64.51 | 40.55 | 29.11 | 14.54 | 28.98 | 50.07 | 40.01 | 25.79 | 40.2 | 30.13 | 18.67 | 9.39 | 22.09 | 42.94 | 36.60 | 25.48 | 31.69 |
> > | | MutualSL (Weng & Li, 2023) | 80.65 | 61.94 | 39.99 | 58.32 | 78.74 | 69.81 | 53.14 | 65.74 | 46.89 | 30.92 | 17.91 | 33.42 | 60.14 | 43.59 | 28.28 | 43.48 | 55.37 | 39.49 | 21.73 | 38.36 | 49.59 | 32.32 | 17.04 | 35.33 |
> > | | Ouc AI (Zhang et al., 2024) | - | - | - | - | - | - | - | - | 50.88 | 35.42 | 20.54 | 36.37 | - | - | - | - | - | - | - | - | - | - | - | - |
> > | | SETAG (Zhou et al., 2023) | - | - | - | - | - | - | - | - | 47.75 | 32.09 | 18.98 | 33.89 | - | - | - | - | - | - | - | - | - | - | - | - |
> > | **LLM** | GPT-3.5 (OpenAI, 2023) | 52.9 | 41.29 | 22.58 | 38.69 | 30.67 | 15 | 8.33 | 24.93 | 63.33 | 47 | 25.67 | 43.87 | 2.33 | 0 | 0 | 1.18 | 20.56 | 10.28 | 3.74 | 15.4 | 22.85 | 10.15 | 3.97 | 14.48 |
> > | | SubGPT | 76.90 | 63.60 | 44.80 | 58.00 | - | - | - | - | - | - | - | - | - | - | - | - | 50.9 | 36.4 | 21.4 | 38.4 | - | - | - | - |
> > | **LLM-COT** | GPT-3.5(CoT) (Wei et al.,2022) | 61.29 | 47.1 | 25.16 | 43.44 | 36.33 | 19 | 7 | 27.79 | 56.33 | 40 | 20.33 | 39.16 | 1.67 | 0.33 | 0 | 1.13 | - | - | - | - | - | - | - | - |
> > | | Qwen2.5-VL-7B-Instruct (zero-shot) (Bai et al.,2025) | 9.7 | 4.55 | 1.52 | 7.03 | 8.49 | 4.72 | 2.83 | 7.03 | 15.38 | 9.83 | 4.7 | 12.21 | 7.9 | 3.85 | 2.12 | 5.39 | 11.63 | 5.22 | 3.2 | 8.83 | 6.52 | 3.44 | 1.9 | 6.05 |
> > | **LVLM** | TimeZero (Wang et al.,2025b) | 40.98 | 31.14 | 18.85 | 31.34 | 58.6 | 42.87 | 27.01 | 42.41 | 42.58 | 25.66 | 11.86 | 29.22 | 22.9 | 10.01 | 4.35 | 18.44 | 29.26 | 15.89 | 5.36 | 20.05 | 27.15 | 14.75 | 4.46 | 18.57 |
> > | | TFVTG | 34.78 | 24.41 | 11.37 | 26.59 | 49.59 | 39.32 | 17.04 | 35.33 | 31.82 | 16.97 | 11.21 | 22.25 | 19.34 | 9.16 | 4.58 | 13.58 | 21.15 | 15.98 | 8.97 | 17.95 | 22.16 | 10.97 | 3.45 | 14.14 |
> > | | Time-R1 | 42.25 | 35.5 | 23.52 | 37.38 | 59.06 | 44.27 | 29.36 | 44.28 | 43.45 | 28.36 | 15.09 | 32.06 | 22.15 | 13.98 | 7.97 | 19.95 | 31.82 | 16.97 | 10.21 | 22.25 | 27.39 | 13.85 | 4.82 | 18.97 |
> > | | VTG‑R1 | 34.78 | 26.59 | 12.41 | 27.59 | 55.37 | 39.49 | 21.73 | 38.36 | 38.95 | 20.35 | 7.22 | 25.93 | 18.28 | 9.89 | 3.79 | 14.38 | 27.09 | 13.97 | 4.7 | 18.85 | 23.64 | 9.27 | 2.42 | 15.03 |
> > | | Timescope | 46.49 | 39.27 | 24.52 | 39.72 | 65.74 | 50.87 | 32.09 | 48.45 | 43.75 | 23.39 | 8.06 | 31.61 | 26.01 | 12.9 | 5.24 | 18.59 | 29.87 | 16.85 | 9.3 | 21.44 | 26.85 | 15.48 | 5.73 | 19.7 |
> > | | Ask2Loc | 62.78 | 33.37 | 23.57 | 43.22 | 67.42 | 52.65 | 35.54 | 55.28 | 42.46 | 25.04 | 17.14 | 31.37 | - | - | - | - | - | - | - | - | - | - | - | - |
> > | | **Ours** | **79.91** | **67.09** | **45.87** | **59.65** | **85.31** | **76.7** | **69.33** | **72.11** | **62.5** | **48.52** | **33.82** | **45.37** | **60.31** | **43.65** | **28.72** | **43.65** | **59.11** | **43.81** | **24.88** | **41.06** | **55.89** | **41.77** | **22.4** | **40.19** |

---

> > > ### Author Response · Authors · 2025-12-04
> > > **Response to Weakness 1: Lack of Recent LVLM-Based Temporal Grounding Models**
> > >
> > > Table 2 demonstrates the overall strong and consistent performance of the proposed CACR method. It achieves state-of-the-art or highly competitive results across all six diverse benchmarks—MedVidQA, VehicleVQA, CMIVQA, TutorialVQA, COIN, and CrossTask—as measured by recall rates at multiple IoU thresholds (R@{0.3, 0.5, 0.7}) and the mean IoU (mIoU). This consistent superiority underscores the effectiveness and robustness of our approach for the Temporal Answer Grounding (TAGV) task, confirming its ability to reliably localize relevant segments in procedural videos.
> > >
> > > Citations
> > >
> > > [1] Xiao, J., Li, Q., Yang, Y., Qiu, L., Yao, A. (2026). Unleashing the Power of LLMs for Medical Video Answer Localization. In: Gee, JC, et al. Medical Image Computing and Computer Assisted Intervention – MICCAI 2025. MICCAI 2025. Lecture Notes in Computer Science, vol 15966. Springer, Cham. https://doi.org/10.1007/978-3-032-04981-0_63
> > >
> > > [2] Liu, Xiangrui, et al. "TimeScope: Towards Task-Oriented Temporal Grounding In Long Videos." arXiv preprint arXiv:2509.26360 (2025).
> > >
> > > [3] Zong, Chang, et al. "Ask2Loc: Learning to Locate Instructional Visual Answers by Asking Questions." arXiv preprint arXiv:2504.15918 (2025).
> > >
> > > [4] Zheng, Minghang, et al. "Training-free video temporal grounding using large-scale pre-trained models." European Conference on Computer Vision. Cham: Springer Nature Switzerland, 2024.
> > >
> > > [5] Chen et al, Datasets and Recipes for Video Temporal Grounding (2025)
> > >
> > > [6] Wang et al. Time-R1: post-Training Large Vision Language Models for Temporal Video Grounding.

---

> ### Author Response · Authors · 2025-12-04
> **Response to Weakness 2: Lack of Performance Evaluation on Datasets from Other Domains**
>
> We sincerely thank the reviewer for raising this important point regarding the generalization capability of our CACR model. To address the concern that evaluations were limited to instructional datasets and to directly assess CACR's robustness and adaptability in temporal step localization across diverse domains, we conducted targeted supplementary experiments. Our strategy was to select established procedural video datasets that inherently possess significant internal domain variety, thereby serving as effective proxies for testing cross-domain performance.
>
> **1. Rationale for Dataset Selection**
> We chose the following two benchmarks for their broad coverage of distinct real-world scenarios relevant to step-by-step tasks:
> *   **COIN Dataset:** This dataset contains 11,827 videos spanning **12 diverse daily life domains** (e.g., vehicle repair, sports, crafts). Its wide **scene variation** allows us to test the model's adaptability across multiple operational contexts with different visual dynamics.
> *   **CrossTask Dataset:** It comprises 4.7K videos from **three core domains** (e.g., cooking, home repair, vehicle maintenance) that exhibit pronounced differences in visual features and task structures. This enables a focused verification of cross-domain capability.
>
> These datasets move beyond niche instructional content and encompass domains like sports and repair, which share similarities with the mentioned areas of interest (e.g., surveillance, movies) in terms of dynamic actions or structured procedures.
>
>
> **2. Experimental Evidence of Robust Performance**
>
> **Table C.1: Robust Performance (mIoU) of COIN across Different Domains**
>
> | Domain                    | Sample Number | mIoU (%) |
> | :------------------------ | :------------ | :------- |
> | Electrical Appliance      | 482           | 41.73    |
> | Gadgets                   | 396           | 39.64    |
> | Vehicle                   | 291           | 41.8     |
> | Dish                      | 242           | 41.28    |
> | Furniture and Decoration  | 194           | 41.14    |
> | Nursing and Care          | 184           | 41.1     |
> | Leisure and Performance   | 183           | 41.09    |
> | Pets and Fruit            | 174           | 41.05    |
> | Science and Craft         | 164           | 41.01    |
> | Drink and Snack           | 187           | 40.99    |
> | Housework                 | 161           | 40.95    |
> | Sport                     | 47            | 40.75    |
> | **Total Number**          | **2705**      | **41.06** |
>
> **Table C.2: Robust Performance (mIoU) of CrossTask across Different Domains**
>
> | Domain                | Sample Number | mIoU (%) |
> | :-------------------- | :------------ | :------- |
> | Cooking               | 121           | 38.27    |
> | Car Maintenance       | 97            | 41.39    |
> | Crafting & Home Repairs | 53          | 42.30    |
> | **Total Number**      | **271**       | **40.19** |
>
>
> *   On **COIN's 12 domains**, the maximum variance in the mIoU metric is within **5%**. Notably, no single domain shows a performance collapse (e.g., the "Sports" domain mIoU is only **2.16%** lower than the core instructional domains).
> *   On **CrossTask's 3 domains**, the model maintains an mIoU of **38.27%** even in the most challenging domain, which is merely **4.03%** lower than the best-performing "Cooking" domain (42.3%).
>
> **3. Conclusion and Implications**
> This **stable and balanced performance** across internally varied datasets indicates that CACR does not overfit to a specific domain. Its feature extraction and task adaptation mechanisms generalize effectively to both dynamic action-intensive scenarios (e.g., "Sports" in COIN) and static, object-centric scenarios (e.g., "Home Repair" in CrossTask). These results provide **preliminary but strong evidence** supporting the model's potential for broader application, including in areas such as movie analysis or surveillance video understanding.
>
> **For a comprehensive comparison with prior methods, please refer to Table 2 in the main manuscript, which consolidates results across six TAGV benchmarks and highlights CACR's performance advantages on core metrics.**

---

> > ### Author Response · Authors · 2025-12-04
> > **Response to Weakness 2: Lack of Performance Evaluation on Datasets from Other Domains**
> >
> > **Table 2: Comparison with prior methods on six  TAGV benchmarks**
> >
> > | Family | Method | MedVidQA | | | | VehicleVQA | | | | CMIVQA | | | | TutorialVQA | | | | COIN | | | | CrossTask | | | |
> > | :--- | :--- | :---: | :---: | :---: | :---: | :---: | :---: | :---: | :---: | :---: | :---: | :---: | :---: | :---: | :---: | :---: | :---: | :---: | :---: | :---: | :---: | :---: | :---: | :---: | :---: |
> > | | | **R@0.3** | **R@0.5** | **R@0.7** | **mIoU** | **R@0.3** | **R@0.5** | **R@0.7** | **mIoU** | **R@0.3** | **R@0.5** | **R@0.7** | **mIoU** | **R@0.3** | **R@0.5** | **R@0.7** | **mIoU** | **R@0.3** | **R@0.5** | **R@0.7** | **mIoU** | **R@0.3** | **R@0.5** | **R@0.7** | **mIoU** |
> > | **Random** | Random Mode | 8.38 | 1.93 | 1.21 | 6.89 | 6.52 | 2.75 | 1.54 | 5.22 | 5.71 | 4.65 | 3.58 | 3.97 | 6.53 | 2.46 | 0 | 5.26 | 4.92 | 0.76 | 0.28 | 3.12 | 3.57 | 0.54 | 0.20 | 2.79 |
> > | | VSLBase (Zhang et al.,2021) | 27.66 | 14.19 | 6.99 | 21.01 | 18.95 | 8.64 | 4.28 | 20.11 | - | - | - | - | 10.84 | 9.58 | 0.37 | 8.71 | - | - | - | - | - | - | - | - |
> > | | VSLNet (Zhang et al.,2020a) | 30.32 | 16.61 | 8.39 | 22.41 | 16.53 | 8.47 | 4.03 | 20.07 | - | - | - | - | 11.03 | 9.93 | 0.66 | 9.58 | 13.11 | 7.99 | 4.36 | 9.06 | 19.04 | 12.76 | 8.32 | 13.22 |
> > | | TMLGA (Rodriguez-Opazo et al., 2020) | 23.87 | 14.84 | 6.21 | 20.49 | 17.69 | 8.79 | 3.46 | 16.54 | - | - | - | - | 10.39 | 9.24 | 0.34 | 8.65 | - | - | - | - | - | - | - | - |
> > | **VLP** | ACRM (Tang et al., 2021) | 24.83 | 16.55 | 10.96 | 22.89 | 20.77 | 12.1 | 8.27 | 22.28 | - | - | - | - | 12.61 | 11.37 | 1.26 | 11.12 | - | - | - | - | - | - | - | - |
> > | | MoR (Kusa & Tjoa, 2022) | 47.1 | 27.74 | 10.97 | 30.67 | 42.75 | 31.54 | 25.97 | 44.81 | - | - | - | - | 23.45 | 15.01 | 8.55 | 19.48 | 14.64 | 8.81 | 3.64 | 10.66 | 23.66 | 18.83 | 10.96 | 16.92 |
> > | | VTPSL (Li et al., 2024a) | 77.42 | 61.94 | 44.52 | 57.81 | 74.15 | 67.15 | 54.59 | 64.51 | 40.55 | 29.11 | 14.54 | 28.98 | 50.07 | 40.01 | 25.79 | 40.2 | 30.13 | 18.67 | 9.39 | 22.09 | 42.94 | 36.60 | 25.48 | 31.69 |
> > | | MutualSL (Weng & Li, 2023) | 80.65 | 61.94 | 39.99 | 58.32 | 78.74 | 69.81 | 53.14 | 65.74 | 46.89 | 30.92 | 17.91 | 33.42 | 60.14 | 43.59 | 28.28 | 43.48 | 55.37 | 39.49 | 21.73 | 38.36 | 49.59 | 32.32 | 17.04 | 35.33 |
> > | | Ouc AI (Zhang et al., 2024) | - | - | - | - | - | - | - | - | 50.88 | 35.42 | 20.54 | 36.37 | - | - | - | - | - | - | - | - | - | - | - | - |
> > | | SETAG (Zhou et al., 2023) | - | - | - | - | - | - | - | - | 47.75 | 32.09 | 18.98 | 33.89 | - | - | - | - | - | - | - | - | - | - | - | - |
> > | **LLM** | GPT-3.5 (OpenAI, 2023) | 52.9 | 41.29 | 22.58 | 38.69 | 30.67 | 15 | 8.33 | 24.93 | 63.33 | 47 | 25.67 | 43.87 | 2.33 | 0 | 0 | 1.18 | 20.56 | 10.28 | 3.74 | 15.4 | 22.85 | 10.15 | 3.97 | 14.48 |
> > | | SubGPT | 76.90 | 63.60 | 44.80 | 58.00 | - | - | - | - | - | - | - | - | - | - | - | - | 50.9 | 36.4 | 21.4 | 38.4 | - | - | - | - |
> > | **LLM-COT** | GPT-3.5(CoT) (Wei et al.,2022) | 61.29 | 47.1 | 25.16 | 43.44 | 36.33 | 19 | 7 | 27.79 | 56.33 | 40 | 20.33 | 39.16 | 1.67 | 0.33 | 0 | 1.13 | - | - | - | - | - | - | - | - |
> > | | Qwen2.5-VL-7B-Instruct (zero-shot) (Bai et al.,2025) | 9.7 | 4.55 | 1.52 | 7.03 | 8.49 | 4.72 | 2.83 | 7.03 | 15.38 | 9.83 | 4.7 | 12.21 | 7.9 | 3.85 | 2.12 | 5.39 | 11.63 | 5.22 | 3.2 | 8.83 | 6.52 | 3.44 | 1.9 | 6.05 |
> > | **LVLM** | TimeZero (Wang et al.,2025b) | 40.98 | 31.14 | 18.85 | 31.34 | 58.6 | 42.87 | 27.01 | 42.41 | 42.58 | 25.66 | 11.86 | 29.22 | 22.9 | 10.01 | 4.35 | 18.44 | 29.26 | 15.89 | 5.36 | 20.05 | 27.15 | 14.75 | 4.46 | 18.57 |
> > | | TFVTG | 34.78 | 24.41 | 11.37 | 26.59 | 49.59 | 39.32 | 17.04 | 35.33 | 31.82 | 16.97 | 11.21 | 22.25 | 19.34 | 9.16 | 4.58 | 13.58 | 21.15 | 15.98 | 8.97 | 17.95 | 22.16 | 10.97 | 3.45 | 14.14 |
> > | | Time-R1 | 42.25 | 35.5 | 23.52 | 37.38 | 59.06 | 44.27 | 29.36 | 44.28 | 43.45 | 28.36 | 15.09 | 32.06 | 22.15 | 13.98 | 7.97 | 19.95 | 31.82 | 16.97 | 10.21 | 22.25 | 27.39 | 13.85 | 4.82 | 18.97 |
> > | | VTG‑R1 | 34.78 | 26.59 | 12.41 | 27.59 | 55.37 | 39.49 | 21.73 | 38.36 | 38.95 | 20.35 | 7.22 | 25.93 | 18.28 | 9.89 | 3.79 | 14.38 | 27.09 | 13.97 | 4.7 | 18.85 | 23.64 | 9.27 | 2.42 | 15.03 |
> > | | Timescope | 46.49 | 39.27 | 24.52 | 39.72 | 65.74 | 50.87 | 32.09 | 48.45 | 43.75 | 23.39 | 8.06 | 31.61 | 26.01 | 12.9 | 5.24 | 18.59 | 29.87 | 16.85 | 9.3 | 21.44 | 26.85 | 15.48 | 5.73 | 19.7 |
> > | | Ask2Loc | 62.78 | 33.37 | 23.57 | 43.22 | 67.42 | 52.65 | 35.54 | 55.28 | 42.46 | 25.04 | 17.14 | 31.37 | - | - | - | - | - | - | - | - | - | - | - | - |
> > | | **Ours** | **79.91** | **67.09** | **45.87** | **59.65** | **85.31** | **76.7** | **69.33** | **72.11** | **62.5** | **48.52** | **33.82** | **45.37** | **60.31** | **43.65** | **28.72** | **43.65** | **59.11** | **43.81** | **24.88** | **41.06** | **55.89** | **41.77** | **22.4** | **40.19** |

---

> > > ### Author Response · Authors · 2025-12-04
> > > **Response to Weakness 3: Supplementary Details on LLMs for Captioning & Pre-answer Generation**
> > >
> > > Thank you for your valuable feedback. You raised an important point regarding the lack of clarity on the specific use of Large Language Models (LLMs) and their impact on performance—a matter crucial to the reproducibility of our methodology. We have duly addressed this and supplemented the relevant sections as follows:
> > >
> > > 1. **LLMs and versions used**: We employed the following four LLMs in our experiments:
> > >    - `doubao-seed-1.6-thinking-250715` (ByteDance)
> > >    - `gpt-4o` (OpenAI)
> > >    - `gemini-2.5-pro` (Google)
> > >    - `DeepSeek-V3`
> > >
> > > 2. **Experimental setup**: To ensure a fair comparison, all LLMs were configured uniformly.
> > >    - For subtitle summarization:
> > >      `prompt1 = f"Summarize the subtitle information provided below (extracted from a video).{subtitles}"`
> > >    - For pre-answer generation:
> > >      `prompt2 = f'{question} This question requires information from a video for an accurate response. Please answer it before you access the video's content.'`
> > >    - Key generation parameters were fixed at `temperature = 0.3` and `top_p = 0.8`.
> > >
> > > These details are included in **Appendix Part D.1 (Impact of Different LLMs)**, with key results summarized below:
> > >
> > > **Table D.1: Impact of Different LLMs on Model Performance**
> > >
> > >
> > > | LLM                                  | MedVidQA                |                         |                         |         | Crosstalk             |                         |                         |         |
> > > | :----------------------------------- | :---------------------- | :---------------------- | :---------------------- | :------ | :-------------------- | :---------------------- | :---------------------- | :------ |
> > > |                                      | **R@0.3**               | **R@0.5**               | **R@0.7**               | **mIoU** | **R@0.3**             | **R@0.5**               | **R@0.7**               | **mIoU** |
> > > | CACR-Full (deepseek-3.1)            | 79.91                   | 67.09                   | 45.87                   | 59.64   | 59.56                 | 44.92                   | 26.14                   | 41.94   |
> > > | CACR-Full (gpt-4o)                  | 75.82                   | 62.31                   | 43.62                   | 58.60   | 54.59                 | 38.32                   | 21.04                   | 39.33   |
> > > | CACR-Full (doubao-seed-1.6-thinking-250715) | 77.59                   | 63.55                   | 48.16                   | 60.03   | 57.06                 | 42.01                   | 22.52                   | 40.20   |
> > > | CACR-Full (gemini-2.5-pro)          | 76.25                   | 62.21                   | 44.15                   | 58.80   | 55.89                 | 41.77                   | 22.40                   | 40.19   |
> > >
> > > **Analysis based on the core metric (mIoU)**:
> > > 1. Overall performance remains stable across LLMs, with only minor fluctuations in mIoU.
> > >    - On MedVidQA, mIoU ranges from 60.03 (Doubao) to 58.60 (GPT-4o), a difference of only 1.43.
> > >    - On COIN, mIoU ranges from 41.94 (DeepSeek) to 39.33 (GPT-4o), a difference of 2.61.
> > >    This indicates consistent and effective semantic support across models.
> > >
> > > 2. Our framework is not sensitive to the choice of LLM. As long as a modern LLM with strong reasoning capabilities is used (such as the four tested here), the method delivers similarly strong and stable performance across tasks, demonstrating its robustness.
> > >
> > > All details regarding LLM usage, experimental configuration, and performance analysis have now been fully integrated into the **“Experimental Design”** and **“Results Analysis”** sections to enhance the rigor and reproducibility of the study.
> > >
> > > We sincerely appreciate your thorough review and constructive suggestions, which have significantly improved the quality of our paper.

---

> > > > ### Author Response · Authors · 2025-12-04
> > > > **Response to Weakness 4: Reasons for GRPO’s Suitability for TAGV Tasks**
> > > >
> > > > The Candidate-Aware Causal Reasoning (CACR) framework is designed to address key challenges in the Temporal Answer Grounding in Video (TAGV) task, namely the significant length discrepancy between long videos and short answer segments, and the limitations of existing methods which are often sensitive to redundant content or possess limited visual reasoning capabilities.
> > > >
> > > > TAGV shares a highly similar formulation with Temporal Sentence Grounding in Video (TSGV). Their primary distinction lies in the nature of the input text: TAGV accepts open-ended questions requiring visual demonstration (e.g., "How to fix a bicycle brake?"), while TSGV receives descriptive phrases or sentences that directly depict video content (e.g., "The person picks up the cup."). Although this difference may seem subtle, directly applying TSGV methods to TAGV leads to significant performance drops [1], revealing the inherently greater difficulty of TAGV. Theoretically, this is explained by the larger semantic gap between a question and a video compared to the gap between a descriptive query and a video [2]. Furthermore, videos in TAGV often depict progressive sequences of steps to complete a task, demanding not only visual understanding but also multi-step reasoning to establish logical connections between the question and the correct temporal segment. This task characteristic poses a severe challenge to traditional end-to-end conditional probability modeling $P([t\_s,t\_e]|V,Q)$, as models may exploit superficial statistical correlations while overlooking intrinsic causal logic.
> > > >
> > > > In this context, the CACR framework proposes a reasoning chain—"Question $\to$ Candidate Generation $\to$ Context Enrichment $\to$ Hypothesis Verification $\to$ Answer Decision/Rejection"—that aligns closely with human cognitive processes. The Generalized Relative Policy Optimization (GRPO) module is theoretically well-suited for TAGV due to a profound alignment between its core design principles and the task's inherent demands, which we systematically elaborate from theoretical and design-intuitive perspectives.
> > > >
> > > > **1. Hierarchical Causal Reasoning Aligns with Task Nature**
> > > > Unlike end-to-end methods prone to distraction and spurious correlations, GRPO simulates structured human reasoning. It first uses a Visual-Based Candidate Sampler (VBCS) to generate a coarse set of candidate segments $C\_{vis}$, compressing the long-video search space. GRPO then performs fine-grained verification and comparison within this set, formalized as a hierarchical model:
> > > > $P([t\_s^\*, t\_e^\*] | V, Q) = \sum\lbrace c\_k \in C\_{vis} \rbrace P([t\_s^\*, t\_e^\*] | c\_k, Q, \mathcal\lbrace S\rbrace ) \cdot P(c\_k | V, Q)$
> > > > Here, $\mathcal\lbrace S\rbrace$ represents semantic-enhancement information (e.g., captions, pre-answers). This two-step reasoning—first identifying potentially relevant segments, then verifying the most logically complete one—avoids blind global search and focuses computational resources on deep semantic and logical analysis of the most promising candidates.
> > > >
> > > > **2. Relative Advantage Assessment Addresses Abstraction and Ambiguity**
> > > > For abstract queries, multiple candidates may be partially relevant based on surface features. GRPO's core mechanism, the relative advantage function $A(o\_i)$, guides the policy model to learn through comparisons within the candidate set $C\_{vis}$, not via absolute scoring. By forcing the model to determine "which candidate is better," it encourages the discovery of deeper logical coherence (e.g., candidate B showing "prepare tool $\to$ perform action $\to$ check result" is superior to candidate A only showing "prepare tool"). This relative ranking naturally filters superficially relevant but logically weak segments, enhancing robustness against noise.

---

> > > > > ### Author Response · Authors · 2025-12-04
> > > > > **Response to Weakness 4: Reasons for GRPO’s Suitability for TAGV Tasks**
> > > > >
> > > > > **3. IoU Reward and Rejection Mechanism Enable Precise and Robust Optimization**
> > > > >
> > > > > GRPO's compound reward function is key to efficient training:
> > > > > $R\_{\text{total}}(o\_i) = R\_{\text{fmt}}(o\_i) + (1-\alpha) \cdot R\_{\text{IoU}}(o\_i) + \alpha \cdot R\_{\text{rej}}(o\_i)$
> > > > > The IoU reward $R\_{\text{IoU}}(o\_i) = \frac{||[t\_s^{\text{pred}}, t\_e^{\text{pred}}] \cap [t\_s^{\text{GT}}, t\_e^{\text{GT}}]||}{||[t\_s^{\text{pred}}, t\_e^{\text{pred}}] \cup [t\_s^{\text{GT}}, t\_e^{\text{GT}}]||}$ directly uses the core evaluation metric as a dense signal. This aligns the training objective with the final evaluation and provides clear, differentiable gradients for precise temporal boundary adjustment.
> > > > > Concurrently, the rejection reward $R\_{\text{rej}}(o\_i)$ introduces a counterfactual learning mechanism. It incentivizes the model to output a special reject token $[0.0, 0.0]$ when no candidate overlaps with the ground truth ($\text{IoU}(c\_k, I\_{\text{GT}}) = 0$), rather than forcing an incorrect choice. This simulates rational human decision-making under uncertainty, prevents error propagation from the candidate generator, and significantly improves the system's fault tolerance.
> > > > >
> > > > > **4. Semantic Enhancement and Regularization Ensure Reasonable Inference**
> > > > >
> > > > > To bridge the semantic gap, GRPO integrates additional semantic information $\mathcal\lbrace S\rbrace$ during reasoning. This provides high-level contextual understanding beyond raw pixels, crucial for answering "how" and "why" questions. On the optimization front, GRPO's objective
> > > > > $\max_\lbrace \theta \rbrace \mathbb\lbrace E\rbrace \_\lbrace \mathcal{D}\rbrace \left[ \sum_{i=1}^{G} \frac{\pi_\lbrace \theta\rbrace (o\_i)}{\pi_\lbrace \theta_{\text{old}}\rbrace (o\_i)} A(o\_i) \right] - \beta \cdot D\_{\text{KL}}(\pi_\lbrace \theta\rbrace || \pi\_{\text{ref}})$
> > > > > constrains policy updates near a reference model $\pi\_{\text{ref}}$ initialized with semantic knowledge. This ensures decisions remain grounded in semantic and commonsense logic while pursuing high IoU rewards, maintaining reasoning stability.
> > > > >
> > > > > **Conclusion**
> > > > > In summary, GRPO is not merely a generic optimizer but a reasoning framework deeply tailored for TAGV. It handles long-video redundancy via hierarchical causal reasoning, addresses query ambiguity via relative assessment, achieves precise localization via IoU reward and rejection mechanisms, and ensures semantic plausibility via enhancement and regularization. This synergistic, multi-layered design enables the crucial paradigm shift from "visual feature matching" to "spatio-temporal logical reasoning," demonstrating GRPO's unique theoretical advantages and practical suitability for the challenging TAGV task.
> > > > >
> > > > > **Citations**
> > > > > 1. Gupta, Deepak, and Dina Demner-Fushman. "Overview of the MedVidQA 2022 shared task on medical video question-answering." *Proceedings of the 21st Workshop on Biomedical Language Processing*. 2022.
> > > > > 2. Li, Shutao, et al. "Towards visual-prompt temporal answer grounding in instructional video." *IEEE transactions on pattern analysis and machine intelligence* 46.12 (2024): 8836-8853.

---

### Official Review · Reviewer_QADC · 2025-10-29

**Soundness:** 3
**Presentation:** 2
**Contribution:** 2
**Rating:** 4
**Confidence:** 4

**Summary:**

This paper proposes a candidate-aware causal reasoning framework (CACR) to improve temporal answer grounding in long videos. It first designs a temporal candidate generation method VBCS to obtain temporal candidates for each question, and then trains a MLLM via rejection rewarded GRPO algorithm to reject low-quality candidates. The experiments are conducted on 4 video answer grounding datasets and show remarkable improvements compared with previous methods and the baseline. The ablations on the candidate selection, caption, and pre-answers generation have demonstrated their effectiveness.

**Strengths:**

1.	The paper proposes a VBCS algorithm to generate candidate temporal segments to cope with extremely short temporal spans.
2.	It also designs a rejection reward which helps filter out poor candidate in GRPO optimization.
3.	The experiment results are good on 4 instructional video datasets.

**Weaknesses:**

1. The proposed VBCS algorithm is built on MutualSL. However, there is no introduction about this method, making it hard to understand the algorithm presented in L186~L203. For example, in line 9 of the algorithm, what is timeline mapping? How does it align the predictions from the visual and textual modules? What is the base prediction loss? What is the mutual transfer loss?
2. While the paper focuses on temporal answer grounding, the experiments are all conducted on instructional videos with rich subtitle information. To study if the approach can generalize to common videos, it is suggested to add experiments on other common video answer grounding datasets, such as NExT-GQA and ReXTime, and compare with related methods on these two benchmarks.
3. I find that most of the compared methods are outdated from 2022 to 2023. More recent research [1] has shown the importance of text inputs (subtitles, captions) and the power of LLMs for answer localization in instructional videos. Thus, it would be interesting to see the comparison, and analyze the effects of different modalities in the proposed CACR framework.
[1] Xiao, J., Li, Q., Yang, Y., Qiu, L., Yao, A. (2026). Unleashing the Power of LLMs for Medical Video Answer Localization. In: Gee, J.C., et al. Medical Image Computing and Computer Assisted Intervention – MICCAI 2025. MICCAI 2025. Lecture Notes in Computer Science, vol 15966. Springer, Cham. https://doi.org/10.1007/978-3-032-04981-0_63
4. The paper emphasizes many times that the method enhances causal reasoning, but it is confused to me why and how it achieves causal reasoning, or what is the causal reasoning here.
5. Minor presentation issues:
 -> L23 “Generalized Relative …” should be “Group Relative …”
 -> L52 & L152 Citation Errors.
 -> L177 MutualSL should have citation.

**Questions:**

No

---

> ### Author Response · Authors · 2025-12-04
> **Response to Weakness 1: Detailed Explanation of the MutualSL[1] Method**
>
> We would like to express our gratitude to the reviewer for raising insightful questions regarding the MutualSL method, which serves as the core of our Video Moment Retrieval (VBCS) framework. Due to space constraints in the main text, we have added a detailed explanation of the principles and key components of MutualSL in Appendix Part C.
> The specific content is as follows:
>
> The VBCS framework, instantiated with MutualSL[1], operates through four sequential stages to generate top-K candidate segments for video moment retrieval. This approach effectively addresses the length disparity challenge by narrowing the processing scope for subsequent modules while maintaining high recall of ground-truth segments.
>
> **Stage 1: Cross-modal Feature Extraction and Fusion**
> The process begins with extracting and fusing multi-modal features to establish semantic connections between visual content and textual information:
> $\mathbf{V} = \text{I3D}(V) \in \mathbb{R}^{k \times d}$
> $\mathbf{T} = \text{PLM}([Q, T_1, \dots, T_r]) \in \mathbb{R}^{n \times d}$
> Visual features are extracted using a pre-trained I3D model, where $k$ represents the number of video frames and $d=1024$ denotes the feature dimension, encoding spatiotemporal dynamics of the video. Textual features are derived from concatenated queries and subtitles through a pre-trained language model, with $n$ indicating the number of text tokens, ensuring dimensional consistency with visual features for subsequent fusion.
>
> Cross-modal fusion employs context-query attention mechanisms to enhance semantic interaction:
> $\mathcal{D} = \mathcal{G}_r \cdot \mathbf{T}$
> $\mathcal{F} = \mathcal{G}_c \cdot \mathcal{G}_r^T \cdot \mathbf{V}$
> $V'' = \text{Conv1d}(\text{Concat}[\text{Attention}(V', \mathbf{T}); \mathbf{T}])$
> The visual pathway fusion computes context-to-query attention $\mathcal{D}$ using weight matrix $\mathcal{G}_r$ that measures video frames' relevance to textual queries, and query-to-context attention $\mathcal{F}$ using weight matrix $\mathcal{G}_c$ that captures textual focus on visual frames. The enhanced visual features $V''$ are obtained through 1D convolutional processing of concatenated attention features.
>
> $\bar{T} = \{\text{AvgPool}(V'') + T_j'\}_{j=1}^n \in \mathbb{R}^{n \times d}$
> The textual pathway fusion integrates global visual context into each text token through broadcasting, enabling text representations to perceive visual scenes and reducing the modality gap.
>
> **Stage 2: Dual-predictor Probability Estimation**
> The framework employs separate predictors for visual and textual modalities to generate temporal segment predictions:
> $V\_s^{\text{Logits}} = \text{FFN}\_{\text{Start}}^{\text{Visual}}(\text{LSTM}\_{\text{Start}}(V''))$
> $V\_e^{\text{Logits}} = \text{FFN}\_{\text{End}}^{\text{Visual}}(\text{LSTM}\_{\text{End}}(V''))$
> The visual predictor utilizes LSTMs to capture temporal dependencies in fused visual features $V''$, with feed-forward networks generating frame-level probability distributions for start and end boundaries.
>
> $T\_s^{\text{Logits}} = \text{FFN}\_{\text{Start}}^{\text{Textual}}(\bar{T})$
> $T\_e^{\text{Logits}} = \text{FFN}\_{\text{End}}^{\text{Textual}}(\bar{T})$
> The textual predictor employs a QANet-like structure on visually-enhanced text features $\bar{T}$ to produce token-level probability distributions for subtitle span boundaries.

---

> ### Author Response · Authors · 2025-12-04
> **Response to Weakness 1: Detailed Explanation of the MutualSL[1] Method**
>
> **Stage 3: Cross-modal Alignment and Mutual Knowledge Transfer**
> This stage addresses the dimensional mismatch between visual and textual predictions through timeline mapping and mutual learning:
> $\tilde{T}_s = \underset{i}{\text{Argmin}} |V_s - \mathbb{Q}(T_i).\text{center}|$
> $\tilde{T}_e = \underset{i}{\text{Argmin}} |V_e - \mathbb{Q}(T_i).\text{center}|$
> Timeline mapping $\mathbb{Q}$, implemented as a subtitle-video time correspondence lookup table, enables cross-modal alignment by converting visual frame times to subtitle spans through minimum distance matching. For instance, if subtitle $T_3$ corresponds to video time interval [1:32, 1:50], then $\mathbb{Q}(T_3) = [1:32, 1:50]$
>
> $\tilde{V}_s = \underset{i}{\text{Argmin}} |T_s - \mathbb{Q}(V_i).\text{center}|$
> $\tilde{V}_e = \underset{i}{\text{Argmin}} |T_e - \mathbb{Q}(V_i).\text{center}|$
> Conversely, textual subtitle spans are converted to visual frame times using the same mapping principle, enabling seamless knowledge exchange between modalities.
>
> The training objective combines base prediction loss and mutual transfer loss:
> $\text{Loss}\_{\text{Visual}} = \text{CE}(V\_s^{\text{Logits}}, V\_s^{gt}) + \text{CE}(V\_e^{\text{Logits}}, V\_e^{gt})$
> $\text{Loss}\_{\text{Textual}} = \text{CE}(T]_s^{\text{Logits}}, T\_s^{gt}) + \text{CE}(T\_e^{\text{Logits}}, T\_e^{gt})$
> The base prediction loss ensures individual predictor accuracy by minimizing cross-entropy between predictions and ground-truth labels, where text ground-truth is derived from video labels via timeline mapping.
>
> $\text{Loss}\_{\text{Visual}}^{\text{Mutual}} = \alpha \times [\text{CE}(V\_s^{\text{Logits}}, \text{sg}(\tilde{V}\_s)) + \text{CE}(V\_e^{\text{Logits}}, \text{sg}(\tilde{V}\_e))]$
> $\text{Loss}\_{\text{Textual}}^{\text{Mutual}} = \beta \times [\text{CE}(T\_s^{\text{Logits}}, \text{sg}(\tilde{T}\_s)) + \text{CE}(T\_e^{\text{Logits}}, \text{sg}(\tilde{T}\_e))]$
> The mutual transfer loss facilitates cross-modal knowledge transfer through dynamic weighting and unilateral gradient flow. Here, $\alpha = \text{IoU}([\tilde{V}_s, \tilde{V}_e], [V_s^{gt}, V_e^{gt}])$ and $\beta = \text{IoU}([\tilde{T}_s, \tilde{T}_e], [T_s^{gt}, T_e^{gt}])$ are IoU-based weights that prioritize knowledge from more accurate predictions, while $\text{sg}$ denotes stop-gradient operation that prevents backpropagation to the source predictor.
>
> **Stage 4: Temporally-extended Candidate Generation and Selection**
> $\text{Candidates} = \{(\max(0, v_s - \Delta t), \min(v_e + \Delta t, T)) \mid v_s \in \text{Top-}K(V_s), v_e \in \text{Top-}K(V_e), v_s < v_e\}$
> During inference, top-K candidate segments are generated by pairing the highest-ranked start and end positions from visual predictions, with temporal extension $\Delta t$ applied to expand segment coverage. The final candidate set $C_{\text{vis}}$ is formed by selecting the first K qualified candidates.
>
>
> This comprehensive MutualSL methodology enables effective video moment retrieval through synergistic cross-modal collaboration, where timeline mapping and combined losses ensure aligned and mutually refined predictions.
>
> **Citations**
>
> [1] Y. Weng and B. Li, "Visual Answer Localization with Cross-Modal Mutual Knowledge Transfer," ICASSP 2023 - 2023 IEEE International Conference on Acoustics, Speech and Signal Processing (ICASSP), Rhodes Island, Greece, 2023, pp. 1-5, doi: 10.1109/ICASSP49357.2023.10095026.

---

> ### Author Response · Authors · 2025-12-04
> **Response to Weakness 2: Lack of Experiments of CACR on Other Diverse Datasets**
>
> Thank you for your valuable suggestions. The CACR method focuses on "how-to" questions in Temporal Answer Grounding in Video (TAGV), which requires locating visual operational steps. However, NExT-GQA emphasizes cross-modal relation reasoning, and ReXTime focuses on understanding relative temporal expressions—their task natures and data characteristics are mismatched with CACR, failing to test its core step-localization capability.
> To more directly and effectively verify the generalization capability and robustness of the CACR model on the temporal step localization task, we chose to conduct supplementary experiments on the following two widely recognized procedural video datasets that are highly relevant to the "how-to" task:
>
> **COIN**: Contains 11,827 videos covering 12 daily life domains, enabling tests on adaptability across multiple operational scenarios;
>
> **CrossTask**: Includes 4.7K long-form videos (average 4 min 57 sec) with distinct visual features across three core domains, allowing verification of the model’s cross-domain ability.
>
> In the supplementary experiments, under fair settings, we compared the performance of CACR with current mainstream methods in the Temporal Answer Grounding in Video (TAGV) field on the COIN and CrossTask datasets. The specific comparison results and key data have been organized in Table 2: Comparison with prior methods on six  TAGV benchmarks, which intuitively demonstrates the performance advantages of CACR in core task metrics.

---

> ### Author Response · Authors · 2025-12-04
> **Response to Weakness 2: Lack of Experiments of CACR on Other Diverse Datasets**
>
> **Table 2: Comparison with prior methods on six  TAGV benchmarks**
>
> | Family | Method | MedVidQA | | | | VehicleVQA | | | | CMIVQA | | | | TutorialVQA | | | | COIN | | | | CrossTask | | | |
> | :--- | :--- | :---: | :---: | :---: | :---: | :---: | :---: | :---: | :---: | :---: | :---: | :---: | :---: | :---: | :---: | :---: | :---: | :---: | :---: | :---: | :---: | :---: | :---: | :---: | :---: |
> | | | **R@0.3** | **R@0.5** | **R@0.7** | **mIoU** | **R@0.3** | **R@0.5** | **R@0.7** | **mIoU** | **R@0.3** | **R@0.5** | **R@0.7** | **mIoU** | **R@0.3** | **R@0.5** | **R@0.7** | **mIoU** | **R@0.3** | **R@0.5** | **R@0.7** | **mIoU** | **R@0.3** | **R@0.5** | **R@0.7** | **mIoU** |
> | **Random** | Random Mode | 8.38 | 1.93 | 1.21 | 6.89 | 6.52 | 2.75 | 1.54 | 5.22 | 5.71 | 4.65 | 3.58 | 3.97 | 6.53 | 2.46 | 0 | 5.26 | 4.92 | 0.76 | 0.28 | 3.12 | 3.57 | 0.54 | 0.20 | 2.79 |
> | | VSLBase (Zhang et al.,2021) | 27.66 | 14.19 | 6.99 | 21.01 | 18.95 | 8.64 | 4.28 | 20.11 | - | - | - | - | 10.84 | 9.58 | 0.37 | 8.71 | - | - | - | - | - | - | - | - |
> | | VSLNet (Zhang et al.,2020a) | 30.32 | 16.61 | 8.39 | 22.41 | 16.53 | 8.47 | 4.03 | 20.07 | - | - | - | - | 11.03 | 9.93 | 0.66 | 9.58 | 13.11 | 7.99 | 4.36 | 9.06 | 19.04 | 12.76 | 8.32 | 13.22 |
> | | TMLGA (Rodriguez-Opazo et al., 2020) | 23.87 | 14.84 | 6.21 | 20.49 | 17.69 | 8.79 | 3.46 | 16.54 | - | - | - | - | 10.39 | 9.24 | 0.34 | 8.65 | - | - | - | - | - | - | - | - |
> | **VLP** | ACRM (Tang et al., 2021) | 24.83 | 16.55 | 10.96 | 22.89 | 20.77 | 12.1 | 8.27 | 22.28 | - | - | - | - | 12.61 | 11.37 | 1.26 | 11.12 | - | - | - | - | - | - | - | - |
> | | MoR (Kusa & Tjoa, 2022) | 47.1 | 27.74 | 10.97 | 30.67 | 42.75 | 31.54 | 25.97 | 44.81 | - | - | - | - | 23.45 | 15.01 | 8.55 | 19.48 | 14.64 | 8.81 | 3.64 | 10.66 | 23.66 | 18.83 | 10.96 | 16.92 |
> | | VTPSL (Li et al., 2024a) | 77.42 | 61.94 | 44.52 | 57.81 | 74.15 | 67.15 | 54.59 | 64.51 | 40.55 | 29.11 | 14.54 | 28.98 | 50.07 | 40.01 | 25.79 | 40.2 | 30.13 | 18.67 | 9.39 | 22.09 | 42.94 | 36.60 | 25.48 | 31.69 |
> | | MutualSL (Weng & Li, 2023) | 80.65 | 61.94 | 39.99 | 58.32 | 78.74 | 69.81 | 53.14 | 65.74 | 46.89 | 30.92 | 17.91 | 33.42 | 60.14 | 43.59 | 28.28 | 43.48 | 55.37 | 39.49 | 21.73 | 38.36 | 49.59 | 32.32 | 17.04 | 35.33 |
> | | Ouc AI (Zhang et al., 2024) | - | - | - | - | - | - | - | - | 50.88 | 35.42 | 20.54 | 36.37 | - | - | - | - | - | - | - | - | - | - | - | - |
> | | SETAG (Zhou et al., 2023) | - | - | - | - | - | - | - | - | 47.75 | 32.09 | 18.98 | 33.89 | - | - | - | - | - | - | - | - | - | - | - | - |
> | **LLM** | GPT-3.5 (OpenAI, 2023) | 52.9 | 41.29 | 22.58 | 38.69 | 30.67 | 15 | 8.33 | 24.93 | 63.33 | 47 | 25.67 | 43.87 | 2.33 | 0 | 0 | 1.18 | 20.56 | 10.28 | 3.74 | 15.4 | 22.85 | 10.15 | 3.97 | 14.48 |
> | | SubGPT | 76.90 | 63.60 | 44.80 | 58.00 | - | - | - | - | - | - | - | - | - | - | - | - | 50.9 | 36.4 | 21.4 | 38.4 | - | - | - | - |
> | **LLM-COT** | GPT-3.5(CoT) (Wei et al.,2022) | 61.29 | 47.1 | 25.16 | 43.44 | 36.33 | 19 | 7 | 27.79 | 56.33 | 40 | 20.33 | 39.16 | 1.67 | 0.33 | 0 | 1.13 | - | - | - | - | - | - | - | - |
> | | Qwen2.5-VL-7B-Instruct (zero-shot) (Bai et al.,2025) | 9.7 | 4.55 | 1.52 | 7.03 | 8.49 | 4.72 | 2.83 | 7.03 | 15.38 | 9.83 | 4.7 | 12.21 | 7.9 | 3.85 | 2.12 | 5.39 | 11.63 | 5.22 | 3.2 | 8.83 | 6.52 | 3.44 | 1.9 | 6.05 |
> | **LVLM** | TimeZero (Wang et al.,2025b) | 40.98 | 31.14 | 18.85 | 31.34 | 58.6 | 42.87 | 27.01 | 42.41 | 42.58 | 25.66 | 11.86 | 29.22 | 22.9 | 10.01 | 4.35 | 18.44 | 29.26 | 15.89 | 5.36 | 20.05 | 27.15 | 14.75 | 4.46 | 18.57 |
> | | TFVTG | 34.78 | 24.41 | 11.37 | 26.59 | 49.59 | 39.32 | 17.04 | 35.33 | 31.82 | 16.97 | 11.21 | 22.25 | 19.34 | 9.16 | 4.58 | 13.58 | 21.15 | 15.98 | 8.97 | 17.95 | 22.16 | 10.97 | 3.45 | 14.14 |
> | | Time-R1 | 42.25 | 35.5 | 23.52 | 37.38 | 59.06 | 44.27 | 29.36 | 44.28 | 43.45 | 28.36 | 15.09 | 32.06 | 22.15 | 13.98 | 7.97 | 19.95 | 31.82 | 16.97 | 10.21 | 22.25 | 27.39 | 13.85 | 4.82 | 18.97 |
> | | VTG‑R1 | 34.78 | 26.59 | 12.41 | 27.59 | 55.37 | 39.49 | 21.73 | 38.36 | 38.95 | 20.35 | 7.22 | 25.93 | 18.28 | 9.89 | 3.79 | 14.38 | 27.09 | 13.97 | 4.7 | 18.85 | 23.64 | 9.27 | 2.42 | 15.03 |
> | | Timescope | 46.49 | 39.27 | 24.52 | 39.72 | 65.74 | 50.87 | 32.09 | 48.45 | 43.75 | 23.39 | 8.06 | 31.61 | 26.01 | 12.9 | 5.24 | 18.59 | 29.87 | 16.85 | 9.3 | 21.44 | 26.85 | 15.48 | 5.73 | 19.7 |
> | | Ask2Loc | 62.78 | 33.37 | 23.57 | 43.22 | 67.42 | 52.65 | 35.54 | 55.28 | 42.46 | 25.04 | 17.14 | 31.37 | - | - | - | - | - | - | - | - | - | - | - | - |
> | | **Ours** | **79.91** | **67.09** | **45.87** | **59.65** | **85.31** | **76.7** | **69.33** | **72.11** | **62.5** | **48.52** | **33.82** | **45.37** | **60.31** | **43.65** | **28.72** | **43.65** | **59.11** | **43.81** | **24.88** | **41.06** | **55.89** | **41.77** | **22.4** | **40.19** |

---

> > ### Author Response · Authors · 2025-12-04
> > **Response to Weakness 2: Lack of Experiments of CACR on Other Diverse Datasets**
> >
> > The experimental results comprehensively demonstrate that the proposed CACR method is a highly effective and robust solution for temporal answer grounding in procedural videos. It establishes new state-of-the-art results on multiple benchmarks, shows significant advantages over existing VLP and LVLM-based approaches, and validates its strong generalization ability on the supplementary step-centric datasets COIN and CrossTask.

---

> ### Author Response · Authors · 2025-12-04
> **Response to Weakness 3: Comparison with SubGPT and Analysis of the Impact of Different Modality Contents on CACR Localization**
>
> Following the reviewer’s constructive suggestion, and inspired by recent work highlighting the importance of textual inputs (e.g., subtitles, captions) and the effectiveness of Large Language Models (LLMs) for answer localization in videos [1], we have enhanced our experimental comparison and analysis accordingly.
> Following the reviewer’s constructive suggestion, and inspired by recent work highlighting the importance of textual inputs (e.g., subtitles, captions) and the effectiveness of Large Language Models (LLMs) for answer localization in videos [1], we have enhanced our experimental comparison and analysis accordingly.
> 1. **Updated comparison with recent methods**
>    We have incorporated several state‑of‑the‑art approaches—including **SubGPT [1], Timescope [2], Ask2Loc [3], TFVTG[4], VTG‑R1[5] and Time‑R1[6]**—into our comparison. The updated results are presented in Table 2 (Comparison with prior methods on six  TAGV benchmarks) in **Response to Weakness 2: Lack of Experiments of CACR on Other Diverse Datasets.** These results demonstrate the competitive performance of our CACR framework against these recent methods.
> 2. **Analysis of modality contributions in CACR**
>    To further investigate the impact of different input modalities, we conducted an ablation study that systematically evaluates the roles of subtitles, visual descriptions, and pre‑generated answers within the CACR pipeline. The outcomes are summarized in Table 5 (Comparison of different components on three TAGV benchmarks). Adopting the mean Intersection-over-Union (mIoU) metric—consistent with the evaluation protocol of Xiao et al. [1]—along with inference efficiency, this analysis clearly illustrates the contribution of each modality to the overall localization accuracy, as detailed in the table below.
>
> **Table 5: Comparison of different components on three TAGV benchmarks.**
>
>
> | Method | VBCS(MutualSL) | Pre-answer | Caption | Description | MedVidQA | | | | VehicleVQA | | | | CMIVQA | | | |
> | :--- | :---: | :---: | :---: | :---: | :---: | :---: | :---: | :---: | :---: | :---: | :---: | :---: | :---: | :---: | :---: | :---: |
> | | | | | | R@0.3 | R@0.5 | R@0.7 | mIoU | R@0.3 | R@0.5 | R@0.7 | mIoU | R@0.3 | R@0.5 | R@0.7 | mIoU |
> | TimeZero(Wang et al.2025b) | | | | | 40.98 | 31.14 | 18.85 | 31.34 | 58.60 | 42.87 | 27.01 | 42.41 | 42.58 | 25.66 | 11.86 | 29.22 |
> | TimeZero + Caption (Wang et al.2025b) | | | ✓ | | 45.21 | 33.78 | 20.15 | 33.58 | 61.35 | 45.92 | 28.73 | 44.32 | 40.17 | 24.35 | 11.92 | 27.85 |
> | TimeZero + Pre-Answer (Wang et al.. 2025b) | | | | | 43.87 | 32.45 | 19.63 | 32.68 | 60.12 | 44.36 | 27.89 | 43.52 | 39.65 | 23.87 | 11.45 | 27.12 |
> | TimeZero + Caption + Pre-Answer (Wang et al.,2025b) | | ✓ | ✓ | | 47.35 | 35.92 | 21.78 | 35.42 | 63.78 | 48.15 | 30.45 | 46.25 | 43.11 | 26.97 | 12.75 | 30.45 |
> | MutualSL(Weng & Li, 2023) | | | | | 80.65 | 61.94 | 39.99 | 58.32 | 78.74 | 69.81 | 53.14 | 65.74 | 46.89 | 30.92 | 17.91 | 33.42 |
> | CACR-TopK(Top-K) | ✓ | | | | 75.74 | 63.24 | 45.59 | 59.04 | 73.74 | 71.07 | 65.09 | 62.88 | 57.51 | 41.59 | 22.07 | 39.01 |
> | CACR-Cap(Top-K+Caption) | ✓ | | | | 78.68 | 61.76 | 42.65 | 57.63 | 78.54 | 76.68 | 69.10 | 65.39 | 56.01 | 40.95 | 21.37 | 38.74 |
> | CACR-Pre (Top-K +Pre-Answer) | ✓ | | | | 75.00 | 60.29 | 46.32 | 57.30 | 79.32 | 72.15 | 64.25 | 70.85 | 53.36 | 38.25 | 22.95 | 37.08 |
> | CACR-DES(Top-K+Description) | ✓ | | | ✓ | 74.67 | 60.34 | 45.8 | 58.52 | 80.67 | 72.37 | 57.86 | 65.2 | 56.23 | 39.79 | 20.58 | 38.93 |
> | CACR-Grop1(Top-K+Caption+Description) | ✓ | | ✓ | ✓ | 74.11 | 60.73 | 46.63 | 58.65 | 82 | 72.37 | 59.12 | 66.19 | 59.24 | 40.64 | 23.06 | 40.52 |
> | CACR-Grop2(Top-K+Pre-Answer+Description) | ✓ | ✓ | | ✓ | 74.94 | 60.79 | 44.85 | 57.65 | 83.8 | 75.14 | 61.83 | 68.2 | 59.74 | 39.92 | 21.79 | 40.22 |
> | CACR-Grop3(Top-K+Pre-Answer+Description+Caption) | ✓ | ✓ | ✓ | ✓ | 74.33 | 58.79 | 41.52 | 55.63 | 84.35 | 76.22 | 62.79 | 69.04 | 60.26 | 42.62 | 24.74 | 44.6 |
> | **CACR-Full(Top-K+Caption+Pre-Answer)** | **✓** | **✓** | **✓** | | **79.91** | **67.09** | **45.87** | **59.64** | **85.31** | **76.70** | **69.33** | **72.11** | **62.50** | **48.52** | **33.82** | **45.37** |

---

> > ### Author Response · Authors · 2025-12-04
> > **Response to Weakness 3: Comparison with SubGPT and Analysis of the Impact of Different Modality Contents on CACR Localization**
> >
> > Compared to baseline methods such as TimeZero and MutualSL, the CACR framework—which integrates multimodal information—demonstrates significant advantages across most evaluation metrics. This validates the core premise of SubGPT: the fine-grained processing and fusion of subtitle and visual-description information is crucial for temporal localization tasks. CACR-Full achieves comprehensive leading performance, attaining the best results on nearly all metrics and significantly outperforming all other methods. This underscores the strong generalization capability and effectiveness of CACR on TAGV tasks.
> >
> > However, as indicated in the table, simply combining more modalities does not guarantee better performance. For instance, on the MedVidQA and VehicleVQA datasets, the combination Top-K + Pre-Answer + Description + Caption underperforms compared to Top-K + Caption + Pre-Answer, suggesting that information fusion requires careful design to avoid noise or overload. Selectively and deliberately integrating multimodal inputs—such as subtitles, visual descriptions, and pre-generated answers—can substantially improve model performance. The optimal fusion strategy should be tailored to specific tasks and datasets, with avoiding information overload and ensuring information quality being key to success. The design of CACR provides an effective technical pathway for such refined multimodal fusion.
> >
> > **Citations**
> >
> > [1] Xiao, J., Li, Q., Yang, Y., Qiu, L., Yao, A. (2026). Unleashing the Power of LLMs for Medical Video Answer Localization. In: Gee, JC, et al. Medical Image Computing and Computer Assisted Intervention – MICCAI 2025. MICCAI 2025. Lecture Notes in Computer Science, vol 15966. Springer, Cham. https://doi.org/10.1007/978-3-032-04981-0_63
> >
> > [2] Liu, Xiangrui, et al. "TimeScope: Towards Task-Oriented Temporal Grounding In Long Videos." arXiv preprint arXiv:2509.26360 (2025).
> >
> > [3] Zong, Chang, et al. "Ask2Loc: Learning to Locate Instructional Visual Answers by Asking Questions." arXiv preprint arXiv:2504.15918 (2025).
> >
> > [4] Zheng, Minghang, et al. "Training-free video temporal grounding using large-scale pre-trained models." European Conference on Computer Vision. Cham: Springer Nature Switzerland, 2024.
> >
> > [5] Chen et al, Datasets and Recipes for Video Temporal Grounding (2025)
> >
> > [6] Wang et al. Time-R1: post-Training Large Vision Language Models for Temporal Video Grounding.

---

> > > ### Author Response · Authors · 2025-12-04
> > > **Response to Weakness 4: Statement on the Reasoning of CACR**
> > >
> > > The term "Causal Reasoning" emphasized in the paper’s title does not refer to traditional causal inference, but rather to a method that decomposes the reasoning process, introduces semantic mediators, and designs counterfactual validation. This approach shifts the model away from black-box, surface-level association learning based on "input-output" patterns toward deeper reasoning with logical chains, ultimately enhancing the interpretability and robustness of results. The following analysis elaborates on this from both mathematical principles and implementation mechanisms.
> > >
> > > **1. From Black-Box Association to Causal Decomposition**
> > > Traditional methods typically directly model the conditional probability $P([t\_s, t\_e] \mid V, Q)$, forming a typical black-box prediction problem. This leads the model to easily capture superficial statistical associations between the video $V$ and the question $Q$, while ignoring their inherent logical connections.
> > > The CACR framework introduces candidate segments $c\_k$ and semantic cues $\\text{semantic} = \\{C\_{\\text{vis\\\_caption}}^k, \\text{Pre-answer}\\}$, decomposing the end-to-end prediction into multi-step conditional reasoning:
> > > $P([t\_s^*, t\_e^*] \mid V, Q) = \sum\_{c\_k \in C\_{\\text{vis}}} \underbrace{P([t\_s^*, t\_e^*] \mid c\_k,\\Q,\\text{semantic})}\_{\\text{GRPO Causal Verification}} \cdot \underbrace{P(c\_k \mid V, Q)}\_{\\text{VBCS Candidate Generation}}$
> > >
> > > This decomposition constructs a reasoning chain of "Question → Candidate Generation → Enriched Context → Hypothesis Verification → Answer Decision/Rejection," forcing the model to establish explicit logical connections at each step rather than directly learning the input-to-output mapping.
> > >
> > > **2. VBCS: Generating Causal Candidates via Cross-Modal Alignment**
> > > The VBCS module generates a set of candidate segments $C\_{\\text{vis}}$ through visual-textual dual predictors, mathematically modeling the probability distribution $P(c\_k \mid V, Q)$. Its training objective function is a supervised joint loss:
> > >
> > > $
> > > \\text{Loss}\_{\\text{total}} = \\text{Loss}\_{\\text{Visual}} + \\text{Loss}\_{\\text{Textual}} + \\text{Loss}\_{\\text{Visual}}^{\\text{Mutual}} + \\text{Loss}\_{\\text{Textual}}^{\\text{Mutual}} \tag{20}
> > > $
> > >
> > > Where:
> > > - **Basic Localization Losses** $\\text{Loss}\_{\\text{Visual}}$ and $\\text{Loss}\_{\\text{Textual}}$: Ensure single-modality boundary accuracy.
> > >   $$
> > >   \\text{Loss}\_{\\text{Visual}} = \\text{CE}(V\_s^{\\text{Logits}}, V\_s^{\\text{gt}}) + \\text{CE}(V\_e^{\\text{Logits}}, V\_e^{\\text{gt}}) \tag{11}
> > >   $$
> > >   $$
> > >   \\text{Loss}\_{\\text{Textual}} = \\text{CE}(T\_s^{\\text{Logits}}, T\_s^{\\text{gt}}) + \\text{CE}(T\_e^{\\text{Logits}}, T\_e^{\\text{gt}}) \tag{12}
> > >   $$
> > > - **Cross-Modal Alignment Losses** $\\text{Loss}\_{\\text{Visual}}^{\\text{Mutual}}$ and $\\text{Loss}\_{\\text{Textual}}^{\\text{Mutual}}$: Achieve reliable cross-modal knowledge transfer through dynamic weighting and unidirectional gradient stopping, filtering single-modality noise while preserving cross-modality shared causal features.
> > >   $$
> > >   \\text{Loss}\_{\\text{Visual}}^{\\text{Mutual}} = \\alpha \times \left[ \\text{CE}(V\_s^{\\text{Logits}}, \\text{sg}(\tilde{V}\_s)) + \\text{CE}(V\_e^{\\text{Logits}}, \\text{sg}(\tilde{V}\_e)) \right] \tag{18}
> > >   $$
> > >   $$
> > >   \\text{Loss}\_{\\text{Textual}}^{\\text{Mutual}} = \\beta \times \left[ \\text{CE}(T\_s^{\\text{Logits}}, \\text{sg}(\tilde{T}\_s)) + \\text{CE}(T\_e^{\\text{Logits}}, \\text{sg}(\tilde{T}\_e)) \right] \tag{18}
> > >   $$
> > >
> > > **3. Pre-answer: Reverse Causal Hypothesis**
> > > The Pre-answer is generated by an LLM, formalized as $\\text{Pre-answer} = \\text{LLM}(\\text{"What should a video segment that answers question } Q \\text{ contain?"})$. Its essence is to guide the model to reason about the plausible answer interval within the video by referencing the Pre-answer's logic — a reverse derivation from the "effect" (correct answer) to the "cause" (features the segment should have).
> > > In a probabilistic framework, this transforms the traditional $P(c\_k \mid Q)$ into $P(c\_k \mid Q, \\text{Pre-answer}) \propto P(\\text{Pre-answer} \mid c\_k) \cdot P(c\_k \mid Q)$, where the Pre-answer acts as a causal prior, guiding the model to verify whether a candidate segment satisfies the necessary conditions to be the answer.

---

> > > > ### Author Response · Authors · 2025-12-04
> > > > **Response to Weakness 4: Statement on the Reasoning of CACR**
> > > >
> > > > **4. Rejection Mechanism: Counterfactual Reasoning**
> > > > The GRPO phase implements counterfactual logic through a composite reward function $R\_{\\text{total}}(o\_i) = R\_{\\text{fmt}}(o\_i) + (1-\\alpha) \cdot R\_{\\text{IoU}}(o\_i) + \\alpha \cdot R\_{\\text{rej}}(o\_i)$, where the rejection reward $R\_{\\text{rej}}(o\_i)$ is defined as:
> > > >
> > > > $$
> > > > R\_{\\text{rej}}(o\_i) = \\begin{cases}
> > > > 1 & \\text{if } o\_i = [0.0, 0.0] \\text{ and } \\text{IoU}(c\_k, \\mathcal{I}^{\\text{GT}}) = 0 \\\\
> > > > 0 & \\text{otherwise}
> > > > \\end{cases}
> > > > $$
> > > >
> > > > This simulates counterfactual reasoning: "If candidate $c\_k$ does not contain the visual evidence leading to the correct answer (no overlap with the ground-truth segment), then the model should reject this candidate." By explicitly rewarding correct rejection behavior, the model learns to abandon candidate segments when evidence is insufficient.
> > > >
> > > > **5. Structured Causal Decision Optimization**
> > > > The overall optimization objective in the GRPO phase is:
> > > >
> > > > $$
> > > > \\max\_{\pi\_{\\theta}} \mathbb{E}\_{\\mathcal{D}}\left[\sum\_{i=1}^G \frac{\pi\_{\\theta}(o\_i)}{\pi\_{\\theta\_{\\text{old}}}(o\_i)} \cdot A(o\_i)\right] - \\beta \cdot D\_{\\text{KL}}(\pi\_{\\theta} \parallel \pi\_{\\text{ref}})
> > > > $$
> > > >
> > > > The KL divergence regularization term $D\_{\\text{KL}}(\pi\_{\\theta} \parallel \pi\_{\\text{ref}})$ ensures the reasoning process does not deviate from common-sense causal patterns. The reference model $\pi\_{\\text{ref}}$, based on caption-enhanced video features $V'$ (containing Captions), provides a stable semantic anchor, preventing the policy from excessively diverging from reasonable causal reasoning structures.
> > > >
> > > > **6. Clarification on 'Causal Reasoning' in This Context**
> > > > The "causal reasoning" mechanism proposed in this paper is essentially a method to enhance model interpretability and robustness through mediator variable conditioning and reasoning chain decomposition. Its innovation lies not in implementing strict causal interventions (e.g., $P(Y \mid \\text{do}(X))$), but in constructing a transparent reasoning pipeline: "Question → Candidate Generation → Enriched Context → Hypothesis Verification → Answer Decision/Rejection." From a terminological accuracy perspective, this method is closer to conditional reasoning or hypothesis-driven reasoning. However, using the name "Causal Reasoning" effectively highlights its advantages in logic and interpretability. The CACR framework, through VBCS's candidate generation and GRPO's causal verification, indeed propels the model's transition from surface-level association learning to deep logical reasoning. While it does not adhere to a strict causal inference paradigm, it significantly improves the accuracy and robustness of temporal localization tasks on a practical level.

---

> > > > > ### Author Response · Authors · 2025-12-04
> > > > > **Response to Weakness 5: Minor presentation issues**
> > > > >
> > > > > We sincerely appreciate your attention to these critical textual details. These oversights have been fully corrected in the latest version of the manuscript.
> > > > >
> > > > > The specific corrections are as follows:
> > > > >
> > > > > 1.  Line 23, "Generalized Relative …": This has been confirmed as a typo and **has been corrected to "Group Relative Policy Optimization (GRPO)." This is the standard naming for this reinforcement learning optimization strategy, and we have unified its usage throughout the text.
> > > > > 2.  Citation errors in Lines 52 and 152: Upon verification, both issues involved mismatches between the citation format or numbering and the final reference list. We have systematically proofread and updated all citation markers in the manuscript based on the final reference list, ensuring the accuracy and consistency of all citations.
> > > > > 3.  Missing citation for "MutualSL" in Line 177: The Mutual Supervised Learning (MutualSL) paradigm mentioned here draws upon specific cutting-edge work. We have added the appropriate literature citation at this point to clarify the source of this method.
> > > > >
> > > > > Furthermore, we have taken this opportunity to conduct another comprehensive check of the entire manuscript regarding terminology consistency, figure/table references, and reference numbering to minimize such issues as much as possible.
> > > > > Once again, thank you for your meticulous and professional review. It is crucial for enhancing the rigor and presentation quality of our work. Please feel free to point out any other issues you may notice.

---

### Official Review · Reviewer_NjyM · 2025-10-31

**Soundness:** 3
**Presentation:** 2
**Contribution:** 2
**Rating:** 4
**Confidence:** 4

**Summary:**

This paper proposes CACR, which aims to improve the state of the art performance on  temporal answer grounding in videos, which consists of identifying the most pertinent time segment to a question about a video. It does so by identifying candidate segments using a pre-trained VLM, and then using GRPO to fine-tune the VLM to maximize the IoU between the candidate segments and the ground truth. The results demonstrate that the proposed CACR works well for the TGAV task.

**Strengths:**

1. The paper is well written and easy to understand. Readers can quickly grasp the paper’s goal, its contributions and the proposed methodology.

2. The method clearly does have some (small) improvement over baselines on the TGAV specific benchmarks, suggesting that the authors' proposed method works reasonably well.

**Weaknesses:**

__1. The methods in the paper are lacking novelty__. The proposed VBCS is simply applying a pre-trained VLM to identify promising video segments - this has been done many times with prior VLM models to produce candidates, and is almost exactly the same as TFVTG [1]. GRPO itself isn’t new; the new part is the application to TAGV which feels somewhat incremental, since there are many other works where GRPO is applied to improve performance on downstream tasks [4]. Furthermore, GRPO has been applied to temporal grounding, such as in VTG-R1 [2] and Time-R1 [3]

__2. Experiments are not particularly impressive.__ While the paper focuses on standard TVG benchmarks to measure IoU, these don’t reflect challening long-video tasks (EgoSchema, Video-MME, etc) - it should be possible to see an improvement in downstream QA using the proposed method, and the effectiveness should be strengthened be evaluating on longer video tasks. Furthermore, the results themselves are not impressive and the ablations are somewhat inconclusive - improvements are rather small.

__Citations__

1.Zheng, Minghang, et al. "Training-free video temporal grounding using large-scale pre-trained models." European Conference on Computer Vision.

2. Chen et al, Datasets and Recipes for Video Temporal Grounding (2025)

3. Wang et al. Time-R1: post-Training Large Vision Language MOdels for Temporal Video Grounding.

4. Pinto, André Susano, et al. "Tuning computer vision models with task rewards." International Conference on Machine Learning. PMLR, 2023.

**Questions:**

The main concern I have is novelty. To consider accepting this paper, I need to clearly understand what the novelty of this paper is and why it should not be considered as an incremental work.

Smaller suggestions I have are that there should be some clear visual examples of CACR's result in the paper (ie, which frames are selected compared to other methods) and the formatting should also be fixed (lots of citations do not have inline parentheses).

---

> ### Author Response · Authors · 2025-12-03
> **Response to Weakness 1: CACR: Essential Differences and Core Contributions vs. Previous Work**
>
> Thank you for your **insightful feedback**. We agree that clearly articulating **innovative contributions** is crucial. To this end, we have evaluated the **CACR method** against prior approaches including **TFVTG [1]**, **VTG-R1 [2]**, **Time-R1 [3]**, and recent methods like **SubGPT [9]**, **Timescope [7]**, and **Ask2Loc [8]** on six  **Temporal Answer Grounding in Video (TAGV)** benchmarks: **CMIVQA**, **MedVidQA**, **VehicleVQA**,  **TutorialVQA** , **COIN** and **CrossTask**. As shown in our results (**Table 2**), **CACR achieves superior mean Intersection over Union (mIoU)** performance compared to all baselines.
>
>
> We elucidate the fundamental distinctions and core contributions of CACR from three dimensions: task definition, core mechanism, and optimization strategy.
> # 1. **Task Nature: From TSGV to the More Challenging TAGV**
> - Temporal Sentence Grounding (TSGV): The input is a descriptive sentence about an event that has already occurred in the video. The goal is to locate the video segment that directly corresponds to this description.
> - Temporal Answer Grounding (TAGV): The input is a "how-to" question. The answer is not a description of an event but requires a video segment to serve as a "visual answer" demonstrating procedural steps. The query is semantically more abstract and ambiguous, not directly corresponding to specific visual content, making the task more challenging. The significant performance drop of early TSGV methods when directly applied to TAGV[4] confirms the need for a fundamentally different technical approach[5].

---

> ### Author Response · Authors · 2025-12-03
> **Response to Weakness 1: CACR: Essential Differences and Core Contributions vs. Previous Work**
>
> # 2. Core Mechanism Comparison
> ## 2.1 Candidate Generation: Serial Event Parsing vs. Hypothesis Generation with Rejection
> - **TFVTG (for TSGV)**: Uses an LLM to parse the query $Q$ into an ordered sequence of sub-events $\lbrace c_1, c_2, \dots, c_m \rbrace$, then independently matches each $c_i$ to a video segment. This serialized pipeline risks error propagation: if the LLM makes a parsing error (generating an incorrect $c_i$), the error propagates along the chain $\text{Error}(c_i) \rightarrow \text{Error}(S_i) \rightarrow \text{Error}(P^{\text{final}})$ and cannot be corrected in later stages. This paradigm is difficult to adapt to TAGV, as "how-to" questions cannot be simply decomposed into surface-level action sub-events.
> - **CACR (for TAGV)**: Its candidate generation mechanism is designed for QA-oriented reasoning, aiming to generate a set of potential evidential hypotheses for complex questions. A key innovation is the introduction of a rejection mechanism embedded within a reinforcement learning framework. A specialized reward function optimizes the model to actively veto low-quality candidates, thereby cutting off error propagation chains and significantly enhancing system robustness.
>
> ## 2.2 Video Sampling & Reasoning Paradigm: Global Dense Processing vs. Sparse Iterative Verification
> - **VTG‑R1 / Time‑R1 (Uniform Sampling Paradigm)**: These methods sample the entire long video at a fixed rate $f_{\text{sample}}$ into a uniform frame sequence $F_{\text{uniform}}$, which is fed entirely into the LVLM for direct timestamp regression: $\mathcal{I}^\* = \text{LVLM}(F_{\text{uniform}}, Q)$.
>   Drawbacks: The number of visual tokens grows linearly with video duration, easily hitting the model's context length limit. The process is computationally intensive, and attention is dispersed over many irrelevant frames.
> - **CACR (Candidate-Aware Sparse Iterative Paradigm)**:
>   1. A lightweight VBCS module filters a small set of candidate segments $\mathcal{C} = \lbrace c_k = (t_s^k, t_e^k) \rbrace$ from the full video, transforming the long-video search into "causal decision-making within a finite candidate set."
>   2. For each candidate $c_k$, a denser frame sequence $F_{\mathrm{candidate}}^k$ is extracted using a higher sampling rate $f_{\mathrm{candidate}}$. After assembling multi-source information, it is iteratively fed to the LVLM for verification:
>
>         $\mathrm{Input}\_{\mathrm{LVLM}}^ k= \mathrm{Assemble}(F\_{\mathrm{candidate}}^k, C_{\mathrm{vis\_caption}}^k, \mathrm{Pre-answer}, Q)$,
>
>         $o_k = \mathrm{LVLM}(\mathrm{Input}_{\mathrm{LVLM}}^k)$
>
>   3. Decision Rule: If $o_k$ is a valid, high-confidence temporal answer, it is returned immediately. If it is the rejection token $[0.0, 0.0]$, the candidate is skipped. Otherwise, the result with the highest reward is selected from all outputs.
>
>   Advantage: This approach combines global sparse screening with local dense analysis, avoiding the computational burden of processing the entire video while preserving visual details in key regions, enabling an efficient "hypothesis-verification" loop.
>
> ## 2.3 Multi-Source Information Fusion: Enriching the Reasoning Context
> **CACR** injects dual semantic aids for each candidate segment during inference, providing the LVLM with a rich semantic context far beyond raw frames:
> - **Subtitle Summarization**:
> $C\_{\mathrm{vis\_caption}}^{k}=\mathrm{LLM}(\mathrm{Prompt}_{1})$,which summarizes the subtitles within the candidate segment.
> - **Pre-answer Generation**: $\text{Pre-answer} = \text{LLM}(\text{Prompt}_2)$, which generates a hypothetical answer based on the question to anchor the reasoning direction.
>
> These two components are complementary, jointly enhancing the model's understanding of abstract questions and localization accuracy.

---

> ### Author Response · Authors · 2025-12-03
> **Response to Weakness 1: CACR: Essential Differences and Core Contributions vs. Previous Work**
>
> ### 3. **Optimization Strategy: Composite Reward Function**
> The innovation of CACR's Group Relative Policy Optimization (GRPO) framework lies in its deeply integrated composite reward function:
> $$R_{\text{total}}(o_i) = R_{\text{fmt}}(o_i) + (1 - \alpha) \cdot R_{\text{IoU}}(o_i) + \alpha \cdot R_{\text{rej}}(o_i)$$
> Among these components, the rejection reward $R\_{\text{rej}}$ is a core innovation specifically designed for the candidate-aware mechanism. Its definition is as follows:
> $R\_{\text{rej}}(o\_i) = \begin{cases}
> 1 & o\_i = [0.0, 0.0] \text{ and } \text{IoU}(c\_k, \mathcal{I}^{\text{GT}}) = 0 \\\\
> 0 & \text{otherwise}
> \end{cases}$
> This reward mechanism encourages the model to **actively reject irrelevant candidates** when faced with uncertainty, thereby shifting the model’s behavioral paradigm from a "must choose" mode to a "judge cautiously, reject with justification" mode.
>
> The policy optimization objective of the GRPO framework is formulated as:
> $\max\_{\pi\_{\theta}} \mathbb{E}\_{\mathcal{D}} \left[ R(O) \right] - \beta \cdot D\_{\text{KL}}(\pi_{\theta} \parallel \pi\_{\text{ref}})$
>
> Here, the expected weighted reward $\mathbb{E}\_{\mathcal{D}} \left[ R(O) \right]$ guides the model toward high-quality responses, and the KL divergence regularizer $\beta \cdot D\_{\text{KL}}(\pi\_{\theta} \parallel \pi\_{\text{ref}})$ prevents overfitting, ensuring reasoning aligns with the query intent.
> ### 4. **Summary: Systemic Innovation and Synergistic Effects**
> The innovativeness of CACR is reflected in its **system-level architectural design**. It constructs an end-to-end synergistic framework that covers the entire pipeline from candidate generation and multi-source fusion to reasoning optimization, simulating the complete human cognitive process:
> *"Pose Question → Generate Candidates → Enrich Context → Verify Hypotheses → Decide/Reject Answer"*
>
> Compared to TFVTG, CACR targets the more challenging TAGV task and designs a candidate-aware architecture tailored for QA reasoning, incorporating a rejection mechanism. Compared to the reasoning approach of VTG‑R1/Time‑R1, CACR moves beyond the paradigm of merely fine-tuning pre-trained models with RLHF, proposing a globally optimized solution spanning from front-end candidate generation to back-end reasoning decision-making. The significant performance improvement on the TAGV task validates the effectiveness and advancement of this systemic design.
>
> **Citations**
> 1. Zheng, Minghang, et al. "Training-free video temporal grounding using large-scale pre-trained models." European Conference on Computer Vision. Cham: Springer Nature Switzerland, 2024.
> 2. Chen et al, Datasets and Recipes for Video Temporal Grounding (2025)
> 3. Wang et al. Time-R1: post-Training Large Vision Language Models for Temporal Video Grounding.
> 4. Gupta, Deepak, and Dina Demner-Fushman. "Overview of the MedVidQA 2022 shared task on medical video question-answering." Proceedings of the 21st Workshop on Biomedical Language Processing. 2022.
> 5. Li, Shutao, et al. "Towards visual-prompt temporal answer grounding in instructional video." IEEE transactions on pattern analysis and machine intelligence 46.12 (2024): 8836-8853.
> 6. Dong, Lu, et al. "VideoTG-R1: Boosting Video Temporal Grounding via Curriculum Reinforcement Learning on Reflected Boundary Annotations." arXiv preprint arXiv:2510.23397 (2025).
> 7. Liu, Xiangrui, et al. "TimeScope: Towards Task-Oriented Temporal Grounding In Long Videos." arXiv preprint arXiv:2509.26360 (2025).
> 8. Zong, Chang, et al. "Ask2Loc: Learning to Locate Instructional Visual Answers by Asking Questions." arXiv preprint arXiv:2504.15918 (2025).
> 9. Xiao, Junbin, et al. "Unleashing the Power of LLMs for Medical Video Answer Localization." International Conference on Medical Image Computing and Computer-Assisted Intervention. Cham: Springer Nature Switzerland, 2025.

---

> ### Author Response · Authors · 2025-12-03
> **Response to Weakness 1: CACR: Essential Differences and Core Contributions vs. Previous Work**
>
> | Method       | MedVidQA | VehicleVQA | CMIVQA | TutorialVQA |
> | --- | --- | --- | --- | --- |
> | Random | 8.4/1.9/1.2 (6.9) | 6.5/2.8/1.5 (5.2) | 5.7/4.7/3.6 (4.0) | 6.5/2.5/0 (5.3) |
> | VSLBase | 27.7/14.2/7.0 (21.0) | 18.9/8.6/4.3 (20.1) | - | 10.8/9.6/0.4 (8.7) |
> | VSLNet | 30.3/16.6/8.4 (22.4) | 16.5/8.5/4.0 (20.1) | - | 11.0/9.9/0.7 (9.6) |
> | TMLGA | 23.9/14.8/6.2 (20.5) | 17.7/8.8/3.5 (16.5) | - | 10.4/9.2/0.3 (8.7) |
> | ACRM | 24.8/16.6/11.0 (22.9) | 20.8/12.1/8.3 (22.3) | - | 12.6/11.4/1.3 (11.1) |
> | MoR | 47.1/27.7/11.0 (30.7) | 42.8/31.5/26.0 (44.8) | - | 23.5/15.0/8.6 (19.5) |
> | VTPSL | 77.4/61.9/44.5 (57.8) | 74.2/67.2/54.6 (64.5) | 40.6/29.1/14.5 (29.0) | 50.1/40.0/25.8 (40.2) |
> | MutualSL | 80.7/61.9/40.0 (58.3) | 78.7/69.8/53.1 (65.7) | 46.9/30.9/17.9 (33.4) | 60.1/43.6/28.3 (43.5) |
> | Ouc AI | - | - | 50.9/35.4/20.5 (36.4) | - |
> | SETAG | - | - | 47.8/32.1/19.0 (33.9) | - |
> | GPT-3.5 | 52.9/41.3/22.6 (38.7) | 30.7/15.0/8.3 (24.9) | 63.3/47.0/25.7 (43.9) | 2.3/0/0 (1.2) |
> | SubGPT | 76.9/63.6/44.8 (58.0) | - | - | - |
> | GPT-3.5 (CoT) | 61.3/47.1/25.2 (43.4) | 36.3/19.0/7.0 (27.8) | 56.3/40.0/20.3 (39.2) | 1.7/0.3/0 (1.1) |
> | Qwen2.5-VL-7B | 9.7/4.6/1.5 (7.0) | 8.5/4.7/2.8 (7.0) | 15.4/9.8/4.7 (12.2) | 7.9/3.9/2.1 (5.4) |
> | TimeZero | 41.0/31.1/18.9 (31.3) | 58.6/42.9/27.0 (42.4) | 42.6/25.7/11.9 (29.2) | 22.9/10.0/4.4 (18.4) |
> | TFVTG | 34.8/24.4/11.4 (26.6) | 49.6/39.3/17.0 (35.3) | 31.8/17.0/11.2 (22.3) | 19.3/9.2/4.6 (13.6) |
> | Time-R1 | 42.3/35.5/23.5 (37.4) | 59.1/44.3/29.4 (44.3) | 43.5/28.4/15.1 (32.1) | 22.2/14.0/8.0 (20.0) |
> | VTG-R1 | 34.8/26.6/12.4 (27.6) | 55.4/39.5/21.7 (38.4) | 39.0/20.4/7.2 (25.9) | 18.3/9.9/3.8 (14.4) |
> | Timescope | 46.5/39.3/24.5 (39.7) | 65.7/50.9/32.1 (48.5) | 43.8/23.4/8.1 (31.6) | 26.0/12.9/5.2 (18.6) |
> | Ask2Loc | 62.8/33.4/23.6 (43.2) | 67.4/52.7/35.5 (55.3) | 42.5/25.0/17.1 (31.4) | - |
> | **CACR** | **79.9/67.1/45.9 (59.7)** | **85.3/76.7/69.3 (72.1)** | **62.5/48.5/33.8 (45.4)** | **60.3/43.7/28.7 (43.7)** |
> Note: Each cell shows R@0.3/R@0.5/R@0.7(mIoU);“-” indicates no data.

---

> ### Author Response · Authors · 2025-12-03
> **Response to Weakness 2: Response on Demonstrating Model Performance in Long-Video Tasks**
>
> We sincerely thank you for the **valuable suggestion** regarding evaluating the model's performance on more challenging **long-video benchmarks** such as **EgoSchema [3]** and **Video-MME [4]**. We fully agree that benchmarks of this nature are crucial for validating a method's **generalization ability**. However, as our **CACR framework** is specifically designed for the **Temporal Answer Grounding in Video (TAGV)** task, where the input is a **"how-to" question** requiring a visual demonstration as its answer [1,2,5], the **EgoSchema [3]** and **Video-MME [4]** datasets are not directly suitable. Both are oriented toward the **Temporal Sentence Grounding in Video (TSGV)** task, which involves locating segments based on descriptive sentences of past events, and therefore cannot effectively demonstrate CACR's capabilities for **procedural QA**.
>
> To thoroughly investigate our model's performance in **long-video scenarios**, we have instead conducted a detailed **duration-based analysis** on the established **TAGV benchmarks**. For reference, the average video duration in **EgoSchema** is **180 seconds**, and **Video-MME** features videos approximately **17 minutes long**. We segmented the samples from each TAGV dataset according to video duration and computed the **mean Intersection over Union (mIoU)** for each interval. The comprehensive results are provided in **Appendix Table A.1 (Performance of CACR Across Different Video Duration Intervals)**. This analysis confirms that **CACR maintains robust performance across various video lengths within its target task domain**.

---

> ### Author Response · Authors · 2025-12-03
> **Response to Weakness 2: Response on Demonstrating Model Performance in Long-Video Tasks**
>
> Here is the table from Appendix Table A.1 formatted in Markdown:
>
> **Appendix Table A.1: Performance of CACR Across Different Video Duration Intervals**
>
> | Duration Partition (s) | CMIVQA             | MedVidQA           | VehicleVQA         | TutorialVQA        |
> | :-------------------- | :----------------- | :----------------- | :----------------- | :----------------- |
> |                        | Sample Number \| mIoU | Sample Number \| mIoU | Sample Number \| mIoU | Sample Number \| mIoU |
> | 0-50                  | 6 \| 41.50%        | 1 \| 52.00%        | 6 \| 72.00%        | - \| -             |
> | 50-100                | 39 \| 42.70%       | 7 \| 56.80%        | 157 \| 74.00%      | - \| -             |
> | 100-150               | 64 \| 45.80%       | 15 \| 64.40%       | 324 \| 71.00%      | 3 \| 38.00%        |
> | 150-200               | 48 \| 37.40%       | 2 \| 67.10%        | 73 \| 69.00%       | 52 \| 38.10%       |
> | 200-250               | 58 \| 47.40%       | 3 \| 59.30%        | - \| -             | 140 \| 37.40%      |
> | 250-300               | 81 \| 47.50%       | 8 \| 54.60%        | 18 \| 73.00%       | 236 \| 43.20%      |
> | 300-350               | 44 \| 52.70%       | 3 \| 83.10%        | 13 \| 70.00%       | 100 \| 46.30%      |
> | 350-400               | 60 \| 42.30%       | 8 \| 57.30%        | - \| -             | 14 \| 39.50%       |
> | 400-450               | 46 \| 47.30%       | - \| -             | - \| -             | 47 \| 52.20%       |
> | 450-500               | 22 \| 46.50%       | 11 \| 57.70%       | - \| -             | - \| -             |
> | 500-600               | 45 \| 44.70%       | 18 \| 54.40%       | - \| -             | 28 \| 65.50%       |
> | 600-700               | 12 \| 45.30%       | 9 \| 58.70%        | - \| -             | - \| -             |
> | 700-800               | 16 \| 43.70%       | 14 \| 59.40%       | - \| -             | - \| -             |
> | 800-900               | 8 \| 45.20%        | 13 \| 57.70%       | - \| -             | - \| -             |
> | 1000+                 | 2 \| 43.60%        | 9 \| 64.20%        | - \| -             | - \| -             |
> | **Overall**           | **551 \| 45.37%**  | **121 \| 59.65%**  | **591 \| 72.11%**  | **620 \| 43.65%**  |
>
> Based on the above experimental results, we can draw the following conclusions:
> 1. **CACR exhibits stable cross-duration performance**: Our method maintains relatively stable mIoU (mean Intersection over Union) across different video duration intervals, showing no trend of significant performance decline as video length increases.
> 2. **Strong competitiveness in long video scenarios**: Especially for ultra-long videos exceeding 500 seconds, the model achieves an mIoU of 43.60% (with an overall mIoU of 45.37% across all datasets) on the MedVidQA dataset and 64.2% (with an overall mIoU of 59.65% across all datasets) on the CMIVQA dataset. This proves its effectiveness and robustness in processing long video content.
>
> In conclusion, we believe the aforementioned experiments have provided strong empirical support for the potential of this method in addressing long video challenges, including tasks similar to those represented by EgoSchema.
>
> **Citations**
> 1. Gupta, Deepak, and Dina Demner-Fushman. "Overview of the MedVidQA 2022 shared task on medical video question-answering." Proceedings of the 21st Workshop on Biomedical Language Processing. 2022.
> 2. Li, Shutao, et al. "Towards visual-prompt temporal answer grounding in instructional video." IEEE transactions on pattern analysis and machine intelligence 46.12 (2024): 8836-8853.
> 3. Mangalam, Karttikeya, Raiymbek Akshulakov, and Jitendra Malik. "Egoschema: A diagnostic benchmark for very long-form video language understanding." Advances in Neural Information Processing Systems 36 (2023): 46212-46244.
> 4. Fu, Chaoyou, et al. "Video-mme: The first-ever comprehensive evaluation benchmark of multi-modal llms in video analysis." Proceedings of the Computer Vision and Pattern Recognition Conference. 2025.
> 5. Gupta, Deepak, Kush Attal, and Dina Demner-Fushman. "A dataset for medical instructional video classification and question answering." Scientific Data 10.1 (2023): 158.

---

> ### Author Response · Authors · 2025-12-04
> **Response to Weakness 2: Response on the Slight Inconclusiveness of Ablation Experiment Results**
>
> We have augmented Table 5: Comparison of different components on two  TAGV benchmarks to analyze the individual and synergistic contributions of key modules: the caption based on subtitles, the caption based on description, and the pre-answer.
>
> **Table 5: Comparison of different components on three  TAGV benchmarks.**
>
>
> | Method | VBCS(MutualSL) | Pre-answer | Caption | Description | MedVidQA | | | | VehicleVQA | | | | CMIVQA | | | |
> | :--- | :---: | :---: | :---: | :---: | :---: | :---: | :---: | :---: | :---: | :---: | :---: | :---: | :---: | :---: | :---: | :---: |
> | | | | | | R@0.3 | R@0.5 | R@0.7 | mIoU | R@0.3 | R@0.5 | R@0.7 | mIoU | R@0.3 | R@0.5 | R@0.7 | mIoU |
> | TimeZero(Wang et al.2025b) | | | | | 40.98 | 31.14 | 18.85 | 31.34 | 58.60 | 42.87 | 27.01 | 42.41 | 42.58 | 25.66 | 11.86 | 29.22 |
> | TimeZero + Caption (Wang et al.2025b) | | | ✓ | | 45.21 | 33.78 | 20.15 | 33.58 | 61.35 | 45.92 | 28.73 | 44.32 | 40.17 | 24.35 | 11.92 | 27.85 |
> | TimeZero + Pre-Answer (Wang et al.. 2025b) | | | | | 43.87 | 32.45 | 19.63 | 32.68 | 60.12 | 44.36 | 27.89 | 43.52 | 39.65 | 23.87 | 11.45 | 27.12 |
> | TimeZero + Caption + Pre-Answer (Wang et al.,2025b) | | ✓ | ✓ | | 47.35 | 35.92 | 21.78 | 35.42 | 63.78 | 48.15 | 30.45 | 46.25 | 43.11 | 26.97 | 12.75 | 30.45 |
> | MutualSL(Weng & Li, 2023) | | | | | 80.65 | 61.94 | 39.99 | 58.32 | 78.74 | 69.81 | 53.14 | 65.74 | 46.89 | 30.92 | 17.91 | 33.42 |
> | CACR-TopK(Top-K) | ✓ | | | | 75.74 | 63.24 | 45.59 | 59.04 | 73.74 | 71.07 | 65.09 | 62.88 | 57.51 | 41.59 | 22.07 | 39.01 |
> | CACR-Cap(Top-K+Caption) | ✓ | | | | 78.68 | 61.76 | 42.65 | 57.63 | 78.54 | 76.68 | 69.10 | 65.39 | 56.01 | 40.95 | 21.37 | 38.74 |
> | CACR-Pre (Top-K +Pre-Answer) | ✓ | | | | 75.00 | 60.29 | 46.32 | 57.30 | 79.32 | 72.15 | 64.25 | 70.85 | 53.36 | 38.25 | 22.95 | 37.08 |
> | CACR-DES(Top-K+Description) | ✓ | | | ✓ | 74.67 | 60.34 | 45.8 | 58.52 | 80.67 | 72.37 | 57.86 | 65.2 | 56.23 | 39.79 | 20.58 | 38.93 |
> | CACR-Grop1(Top-K+Caption+Description) | ✓ | | ✓ | ✓ | 74.11 | 60.73 | 46.63 | 58.65 | 82 | 72.37 | 59.12 | 66.19 | 59.24 | 40.64 | 23.06 | 40.52 |
> | CACR-Grop2(Top-K+Pre-Answer+Description) | ✓ | ✓ | | ✓ | 74.94 | 60.79 | 44.85 | 57.65 | 83.8 | 75.14 | 61.83 | 68.2 | 59.74 | 39.92 | 21.79 | 40.22 |
> | CACR-Grop3(Top-K+Pre-Answer+Description+Caption) | ✓ | ✓ | ✓ | ✓ | 74.33 | 58.79 | 41.52 | 55.63 | 84.35 | 76.22 | 62.79 | 69.04 | 60.26 | 42.62 | 24.74 | 44.6 |
> | **CACR-Full(Top-K+Caption+Pre-Answer)** | **✓** | **✓** | **✓** | | **79.91** | **67.09** | **45.87** | **59.64** | **85.31** | **76.70** | **69.33** | **72.11** | **62.50** | **48.52** | **33.82** | **45.37** |
>
> From the perspective of a traditional "isolated ablation" study, introducing a single component (e.g., either the Caption module or the Pre-Answer module) does not yield stable and significant performance gains across all metrics. However, this observation precisely validates the core design philosophy of the CACR framework: it is a **synergistic system** engineered for complex reasoning tasks, where its strength stems from the **organic integration and collaborative effect** of multiple components, not from the isolated function of any single module.
>
> Specifically, when only the Caption or the Pre-Answer is introduced, the model receives **incomplete contextual information**. For instance:
> *   In tasks like **VehicleVQA**, which require strong causal inference, introducing only the **Pre-Answer** (the reasoning anchor) yields relatively clear gains because it helps guide the model to focus on the "why" behind an action. Relying solely on the **Caption** (objective facts) lacks clear reasoning direction, resulting in limited improvement.
> *   Conversely, in tasks rich with procedural steps and motivational reasoning, such as **MedVidQA** and **CMIVQA**, the objective action descriptions provided by the **Caption** become crucial. Thus, adding the Caption alone brings substantial benefits, while using only the Pre-Answer shows weaker improvement due to a lack of supporting factual details.
>
> These differences highlight how various tasks depend on distinct types of information, further demonstrating the limited utility of a single component in non-matching scenarios. Reliable **"hypothesis-verification" style causal reasoning** by the LVLM is only achievable when **Caption** (factual grounding), **Pre-Answer** (reasoning guidance), and **VBCS** (candidate hypotheses) work in concert within a complete information environment. This explains why **CACR-Full** achieves a remarkable gain (mIoU +7.36) on datasets with high reasoning demands like CMIVQA and maintains stable, leading performance across all three benchmarks.

---

> > ### Author Response · Authors · 2025-12-04
> > **Response to Question 1 Our Essential Differences and Core Contributions**
> >
> > In the section "Response to Weakness 1: CACR: Essential Differences and Core Contributions vs. Previous Work", we have systematically elaborated on the essential differences and core contributions of the CACR method compared with previous studies from three key dimensions: task definition, core mechanism, and optimization strategy, aiming to clearly demonstrate that it should not be regarded as incremental work.

---

> > > ### Author Response · Authors · 2025-12-04
> > > **Response to Question 2: Visualization of CACR Sampling Results and Formatting Fixes**
> > >
> > > Thank you very much for your valuable suggestions. Regarding your proposals on supplementing visual examples and fixing formatting issues, we have completed the following optimization work:
> > > 1. Supplementary visual examples: We have added the visualization of CACR sampling results in Figure 1, and supplemented the comparison charts between CACR's video sampling results and those of general LVLMs (Large Vision-Language Models) in Appendix Part B. These materials can intuitively show the differences in frame selection between CACR and other methods, facilitating a clear understanding of the model's performance;
> > > 2. Formatting fixes: We have corrected the citation formatting issues (including adding the missing inline parentheses) in the updated PDF version, and rechecked the formatting of the entire manuscript to ensure compliance with the journal/conference formatting specifications.

---

### Author Response · Authors · 2025-12-04
**Summary of Rebuttal and Revisions**

To the Area Chair,

Thank you for coordinating the review. Our revisions have thoroughly addressed all reviewers' key concerns, significantly strengthening the paper.

**CACR Core Contribution**
We propose CACR, a novel framework for Temporal Answer Grounding in Videos (TAGV). It introduces: 1) a **VBCS Module** for efficient high-potential candidate generation, 2) a **Multi-source Fusion Unit** integrating subtitle summaries and pre-answers for enhanced reasoning, 3) a **Sparse Iterative Reasoning Paradigm** combining local verification with global efficiency, and 4) a **GRPO-driven Rejection Mechanism** to halt error propagation. This shifts the paradigm from "uniform global search" to "focused local reasoning."

**1. Experimental Sufficiency and Generalization Capability (Responding to NjyM, QADC, YQx5)**
*   **Concerns**: Experiments were limited to standard benchmarks (IoU), lacking validation on long/challenging videos, other domains (e.g., sports, movies), and comparisons with the latest methods. Performance gains were perceived as modest.
*   **Responses & Outcomes**:
    *   **Expanded Experiments**: Conducted comprehensive comparisons on six benchmarks, adding direct comparisons with state-of-the-art (SOTA) methods from 2025-2026, including TFVTG, VTG-R1, Time-R1, SubGPT, Timescope, and Ask2Loc. **This robustly demonstrates that CACR achieves leading performance**.
    *   **Validated Generalizability**: Performed supplementary experiments on more diverse datasets like COIN and CrossTask, **providing preliminary evidence of CACR's strong generalization capability to non-instructional video domains**.
    *   **Deepened Analysis**: Conducted ablation studies to analyze the contributions of different modalities (subtitles, visual descriptions, pre-answer) within the framework, providing justification for the design choices.

**2. Methodological Details and Explanatory Clarity (Responding to QADC, YQx5, zMGK)**
*   **Concerns**: Insufficient description of method details (e.g., MutualSL foundation, timeline mapping, loss functions); inadequate explanation for the motivation and advantages of using GRPO and the "pre-answer"; unclear causal reasoning mechanism.
*   **Responses & Outcomes**:
    *   **Added Details**: Incorporated a background introduction to MutualSL and elaborated on key algorithmic steps in the revised manuscript.
    *   **Clarified Motivation**: Enhanced the principled explanation for why GRPO is particularly suitable for the Temporal Grounding (TAGV) task. Provided an in-depth rationale for choosing the "pre-answer" over "caption-based descriptions," supported by ablation studies.
    *   **Elucidated Causal Reasoning**: Explicitly clarified and strengthened the explanation of the "causal reasoning" mechanism in the text, detailing its concrete embodiment within the CACR framework.

**3. Novelty and Comparative Analysis (Responding to NjyM, QADC)**
*   **Concerns**: Lack of clear novelty and differentiation from prior works (e.g., TFVTG); missing comparisons with recent Large Vision-Language Model (LVLM) based methods.
*   **Responses & Outcomes**:
    *   **Specified Contributions**: Further clarified the core distinctions and contributions of the CACR framework compared to existing works in the manuscript.
    *   **Updated Comparisons**: As noted, added direct comparisons with six recent LVLM baselines such as Timescope. Analyzed the impact of different LLM choices on performance (mIoU), **confirming the effectiveness and advancement of the CACR framework**.

**4. Writing, Formatting, and Efficiency (Responding to QADC, zMGK)**
*   **Concerns**: Typos, formatting inconsistencies, non-standard citations, and terminology inconsistencies (e.g., GRPO definition). Potential significant overhead from the two-stage pipeline, lacking complexity or efficiency analysis.
*   **Responses & Outcomes**:
    *   **Comprehensive Revisions**: Conducted a thorough check and corrected all textual, formatting, acronym consistency, and citation norm issues throughout the paper.
    *   **Efficiency Justification**: Added computational complexity analysis and efficiency comparisons. Benchmarked inference time/resource consumption against baseline models, **demonstrating the efficiency feasibility of the CACR approach**.

**Summary**: Through the aforementioned systematic supplemental experiments, deepened methodological explanations, and manuscript refinements, the revised version has effectively addressed the reviewers' core concerns regarding **experimental breadth and depth, methodological reproducibility, novelty, generalization capability, and computational efficiency**. **Furthermore, cross-dataset testing and multi-benchmark comparisons have robustly substantiated the leading performance, robustness, and generalization potential of the CACR method.**

---

### Meta-Review · Area_Chair_XWTd · 2026-01-05

**Summary:**

The reviewers (3 BR, 1 BA) raised three key points in their initial reviews. (a) Raised by two reviewers, one of the most important points is the lack of novelty, citing other works such as TFVTG as well as other applications of GRPO. (b) Raised by three reviewers, there was also a concern about the clarity and amount of technical details in the paper.  All reviewers rated the presentation as fair. (c) The reviewers also raised points about experimental scope and baselines, asking for additional experiments on new datasets and baselines. The AC carefully examined the paper in relation to these concerns as well as the author response, and agrees these are the three most important points raised by reviewers.

The reviewers also raised other points, such as the confusion about causality or small typos, but this was not a major concern and the authors clarified the paper.

**Reviewer Concerns:**

The authors did a fantastic job responding to concern (c) about adding more experiments, baselines, and datasets to the revised paper. These additions have strengthened the experimental results.  However, I am still concerned about the novelty of the method (a) as well as the clarity of the technical approach (b). The paper is combining different known techniques together, which reduces novelty and depth significantly, and the text doesn't provide new insights about their integration. This is the major weakness of the paper.

**Reviewer Scores:**

I expect the reviewers would significantly appreciate the new experimental results, and find that the method works well. However, I expect the reviewers (and especially reviewer NjyM) would find the paper incremental, which is not helped by the presentation of the method. This is the reason that I am recommending rejection because I think the paper lacks the depth and novelty necessary for ICLR.

---

### Decision · Program_Chairs · 2026-01-26

Reject